# Control of SARS-CoV-2 infection by MT1-MMP-mediated shedding of ACE2

Xuanming Guo [1,11], Jianli Cao[2,11], Jian-Piao Cai[3,11], Jiayan Wu[1], Jiangang Huang[4], Pallavi Asthana[1], Sheung Kin Ken Wong[5], Zi-Wei Ye[2], Susma Gurung[1], Yijing Zhang[1], Sheng Wang[6], Zening Wang[7], Xin Ge [7], Hiu Yee Kwan [1], Aiping Lyu [1], Kui Ming Chan [8], Nathalie Wong[9], Jiandong Huang [5], Zhongjun Zhou[5], Zhao-Xiang Bian [10], Shuofeng Yuan [3] ✉ & Hoi Leong Xavier Wong [1] ✉

Severe acute respiratory syndrome coronavirus 2 (SARS-CoV-2) has caused a global pandemic. Angiotensin-converting enzyme 2 (ACE2) is an entry receptor for SARS-CoV-2. The full-length membrane form of ACE2 (memACE2) undergoes ectodomain shedding to generate a shed soluble form (solACE2) that mediates SARS-CoV-2 entry via receptor-mediated endocytosis. Currently, it is not known how the physiological regulation of ACE2 shedding contributes to the etiology of COVID-19 in vivo. The present study identifies Membrane-type 1 Matrix Metalloproteinase (MT1-MMP) as a critical host protease for solACE2-mediated SARS-CoV-2 infection. SARS-CoV-2 infection leads to increased activation of MT1-MMP that is colocalized with ACE2 in human lung epithelium. Mechanistically, MT1-MMP directly cleaves memACE2 at M706-S to release solACE2$_{18-706}$ that binds to the SARS-CoV-2 spike proteins (S), thus facilitating cell entry of SARS-CoV-2. Human solACE2$_{18-706}$ enables SARS-CoV-2 infection in both non-permissive cells and naturally insusceptible C57BL/6 mice. Inhibition of MT1-MMP activities suppresses solACE2-directed entry of SARS-CoV-2 in human organoids and aged mice. Both solACE2 and circulating MT1-MMP are positively correlated in plasma of aged mice and humans. Our findings provide in vivo evidence demonstrating the contribution of ACE2 shedding to the etiology of COVID-19.

The coronavirus disease (COVID-19) is caused by rapid spread of severe acute respiratory syndrome coronavirus 2 (SARS-CoV-2). Previous SARS-CoV outbreaks in 2003 has provided insights into the receptor required by the virus for animal to human transmission. It has been confirmed that angiotensin converting enzyme 2 (ACE2) identified as the receptor for SARS-CoV is also the receptor for SARS-CoV-2[1,2]. The entry of the SARS-CoV-2 is mediated by the binding of spike (S) protein with ACE2[1–4]. Two proteolytic cleavage events of S protein are

[1]School of Chinese Medicine, Hong Kong Baptist University, Hong Kong SAR, China. [2]Department of Microbiology, Li Ka Shing Faculty of Medicine, The University of Hong Kong, Hong Kong SAR, China. [3]State Key Laboratory of Emerging Infectious Diseases, Carol Yu Centre for Infection, Department of Microbiology, School of Clinical Medicine, Li Ka Shing Faculty of Medicine, The University of Hong Kong, Hong Kong SAR, China. [4]Fujian Provincial Key Laboratory of Innovative Drug Target Research and State Key Laboratory of Cellular Stress Biology, School of Pharmaceutical Sciences, Xiamen University, Xiamen, China. [5]School of Biomedical Sciences, Li Ka Shing Faculty of Medicine, The University of Hong Kong, Hong Kong SAR, China. [6]Respiratory Department, Jinhua Guangfu Hospital, Jinhua, China. [7]Institute of Molecular Medicine, University of Texas Health Science Center at Houston, Houston, TX, USA. [8]Department of Biomedical Sciences, City University of Hong Kong, Hong Kong SAR, China. [9]Department of Surgery, The Chinese University of Hong Kong, Prince of Wales Hospital, N.T., Hong Kong SAR, China. [10]Centre for Chinese Herbal Medicine Drug Development Limited, Hong Kong Baptist University, Hong Kong SAR, China. [11]These authors contributed equally: Xuanming Guo, Jianli Cao, Jian-Piao Cai. ✉e-mail: yuansf@hku.hk; xavierwong@hkbu.edu.hk

essential for the SARS-CoV-2 entry process. The first cleavage at the junction of the S1 and S2 units mediated by furin takes place in the infected cells during virus maturation[5,6]. Upon the binding of S protein with ACE2, the S2′ site is further cleaved at the target cells by host proteases that involve transmembrane protease serine 2 (TMPRSS2)[7–9] and cathepsin L[10,11]. In target cells with high expression of TMPRSS2, the S2′ cleavage is performed by TMPRSS2 on the cell surface, facilitating the cell entry of SARS-CoV-2 through plasma membrane fusion[12]. However, if the target cell expresses insufficient TMPRSS2, SARS-CoV-2-ACE2 complex enters target cells through the endocytic pathway in which the viral complex is internalized into the endosomes by clathrin-mediated endocytosis and the S2′ cleavage is mediated by cathepsins[10,11,13,14]. These cleavage events allow the exposure of the fusion domain and subsequent fusion of the viral envelope with the cellular envelope, facilitating viral uncoating and replication[15].

ACE2 proteins exist in two major forms including the full-length membrane form (memACE2) and a shed soluble form (solACE2). Soluble ACE2 is generated by the proteolytic shedding of transmembrane ACE2. Currently, the role of solACE2 in SARS-CoV-2 infection remains to be elucidated. Under the physiological situation, solACE2 released by cell surface proteases serves as an entry receptor for SARS-CoV-2, facilitating receptor-mediated endocytosis of virus[16]. Suppression of solACE2 generation by inhibition of sheddase activities reduces SARS-CoV/SARS-CoV-2 entry[16,17]. On the contrary under the pharmacological dose, clinical grade human solACE2 and recombinant human solACE2 fusion protein (rACE2) can neutralize SARS-CoV-2 infection in human organoid culture and reduce the susceptibility of SARS-CoV-2 infection in transgenic mice expressing human ACE2[18–20]. However, a clinical trial testing the in vivo effect of rACE2 in COVID-19 patient showed that the treatment with high dosage of rACE2 led to a temporary, but significant, increase in the viral load in tracheal aspirations and nasopharyngeal swabs[21], which seems to be contradictory to the observations in vitro and animal studies. Although the patient eventually recovered from SARS-CoV-2 infection, whether the recovery of patient was directly resulted from the treatment of rACE2 remains to be determined. Therefore, there is an ongoing need to identify the mechanism underlying the generation of solACE2 and its optimum physiological levels regulating the SARS-CoV-2 infectivity. As of yet, the major physiological contributors to ACE2 shedding and the generation of soluble ACE2 are still unclear. Identification of the major sheddases for ACE2 would further our mechanistic understandings of SARS-CoV-2 infection process.

Matrix metalloproteinases (MMPs) comprise a family of 24 Zn-containing endopeptidases, responsible for extracellular remodeling and pericellular proteolysis. Membrane Type 1 Matrix Metalloproteinase (MT1-MMP/MMP14), a membrane bound MMP, is one of the most investigated MMPs. Physiologically, it cleaves a wide variety of substrates ranging from the extracellular matrix to growth factor receptors. Among all reported MMPs knockout mouse models, only deficiency in MT1-MMP leads to spontaneous phenotypes including growth retardation and defective bone development[22,23]. In contrast, the upregulation of MT1-MMP contributes to various age-related diseases and conditions, such as diabetes, obesity and increased risk of cancers, highlighting its diverse roles in both physiological and pathological conditions[24–26]. Ectodomain shedding is one of the known mechanisms regulating the expression of ACE2. Our previous findings along with other studies have demonstrated that MT1-MMP plays a major role in the proteolytic events of cell surface proteins[27–29]. It is therefore conceivable to postulate that MT1-MMP may contribute to the ectodomain shedding of ACE2.

## Results

To investigate whether MT1-MMP is involved in the ectodomain shedding of ACE2, we examined the expression of ACE2 in the loss of MT1-MMP. Accordingly, we obtained primary lung epithelial cells from wild-type (WT) and *Mmp14* knockout mice and probed their ACE2 levels, since ACE2 is abundantly expressed in lung epithelial cells that are also a major route of SARS-CoV-2 infection. Western blots against the extracellular domain of ACE2 revealed that the protein expression of ACE2 was slightly upregulated in *Mmp14*[-/-] epithelial cells whereas the level of soluble ACE2 (solACE2) was remarkably reduced in the conditioned media from *Mmp14*[-/-] cells (Fig. 1a). As the transcription of ACE2 was not altered by the loss of MT1-MMP (Fig. 1b), these results suggest that MT1-MMP regulates ACE2 at the protein level. Similar observations on the changes in ACE2 level were found in kidney epithelial cells in which ACE2 was highly expressed (Supplementary Fig. 1a). To further investigate whether our observations obtained from mouse are physiologically relevant to human, HEK293T cells stably expressing human ACE2 were transfected with either empty vector, wild-type or catalytically inactive human MT1-MMP. The ectodomain fragment of ACE2 was only increased significantly in the conditioned media derived from wild-type MT1-MMP transfected cells (Fig. 1c), suggesting that MT1-MMP may shed ACE2 and the catalytic activity of MT1-MMP is essential for ACE2 shedding. To investigate whether other MMPs may be involved in ACE2 shedding, ACE2-expressing 293T cells were transfected with plasmids expressing different major MMPs including MMP2, MMP9 and MMP13 and MT1-MMP. We found that only MT1-MMP could cleave the full length ACE2 and release solACE2 into the conditioned medium (Supplementary Fig. 2a), suggesting that ACE2 shedding is specifically mediated by MT1-MMP. In contrast, knockdown of MT1-MMP effectively suppressed the release of solACE2 from human bronchial epithelial cells and human cardiac organoids formed by iPSC-derived cardiomyocytes, two major cell targets of SARS-CoV-2 (Fig. 1d). Similar suppression of solACE2 release by MT1-MMP knockdown was observed in Huh-7 and Caco2 cells, two human cell lines with high susceptibility to SARS-CoV-2 (Supplementary Fig 1b). To further investigate whether ACE2 is a direct substrate of MT1-MMP, recombinant ACE2 was incubated with the catalytic domain of MT1-MMP (cMT1) in vitro. Full length ACE2 was significantly reduced and truncated fragments of ACE2 were detected in the presence of cMT1, which was inhibited by the potent MT1-MMP inhibitor EDTA (Fig. 1e). To further substantiate our findings, we examined the endogenous interaction between MT1-MMP and ACE2 in Caco2 cells by co-immunoprecipitation (Fig. 1f). MT1-MMP was pulled down in ACE2 immunoprecipitate. Reciprocally, ACE2 was detected in MT1-MMP immunoprecipitate. Similarly, the physical interaction between MT1-MMP and ACE2 was also observed in HEK293T cells with ectopic expression of human MT1-MMP and human ACE2 (Supplementary Fig. 2b). To further confirm MT1-MMP directly interacts with ACE2, we performed co-immunoprecipitation using recombinant ACE2 (rACE2) and MT1-MMP protein derived from 293T cells with ectopic expression of MT1-MMP (Supplementary Fig. 2c). Consistently, ACE2 immunoprecipitate could pull down MT1-MMP. The putative MT1-MMP cleavage sites of ACE2 were predicted by Cleavpredict software, a useful platform for reasoning the proteolytic events of MMPs[30]. Based on the predicted cleavage sites with the highest matching scores, we generated a series of ACE2 mutants by site-directed mutagenesis. We found that only mutation at S707A eliminated solACE2 released into the conditioned media of 293T cells co-expressing MT1-MMP and different ACE2 mutants (Supplementary Fig. 3a), indicating that M706-S is likely the physiological MT1-MMP cleavage site of ACE2. Notably, the membrane localization of ACE2 and the physical interaction between MT1-MMP and ACE2 were not altered by S707A mutation (Supplementary Fig. 3b, c), suggesting that the blockage of ACE2 cleavage by S707A mutation is directly resulted from the loss of MT1-MMP cleavage site. To further confirm the identified cleavage site, the truncated fragment of ACE2 resulted from MT1-MMP cleavage in the in vitro cleavage assay was sequenced by mass spectrometry (MS) coupled with tandem MS/MS, revealing that MT1-MMP indeed cleaved C-terminus of ACE2 at M706-S (Supplementary Fig. 4). To further substantiate our

sequencing results, two synthetic ACE2 polypeptides with the identified cleavage site including ACE2$_{691-710}$ and ACE2$_{701-720}$ were digested with the recombinant catalytic domain of MT1-MMP. MS analyses followed by tandem MS/MS revealed that both peptides were consistently cleaved at M706-S (Supplementary Fig. 5). These data collectively suggest that MT1-MMP directly cleaves memACE2 at M706-S to release solACE2 from the cell surface (Fig. 1g).

To further investigate the physiological relevance of ACE2 regulation by MT1-MMP, we analyzed rich reference datasets that describe the single-cell transcriptome of the lung in both healthy individuals and patients infected with SARS-CoV-2[31–33]. By analyzing single-cell transcriptomic data from the Human Protein Atlas[34], we found that *MMP14* was expressed broadly in the lung epithelium and highly expressed in both alveolar type 1 (AT1) and AT2 cells (Fig. 2a, b). Previous studies reveal that the expression of ACE2 is restricted to epithelial cells, especially for secretory cells and a subpopulation of alveolar type 2 (AT2) cells, a primary cell target of SARS-CoV-2[35–38], suggesting that the expression patterns of MT1-MMP and ACE2 are

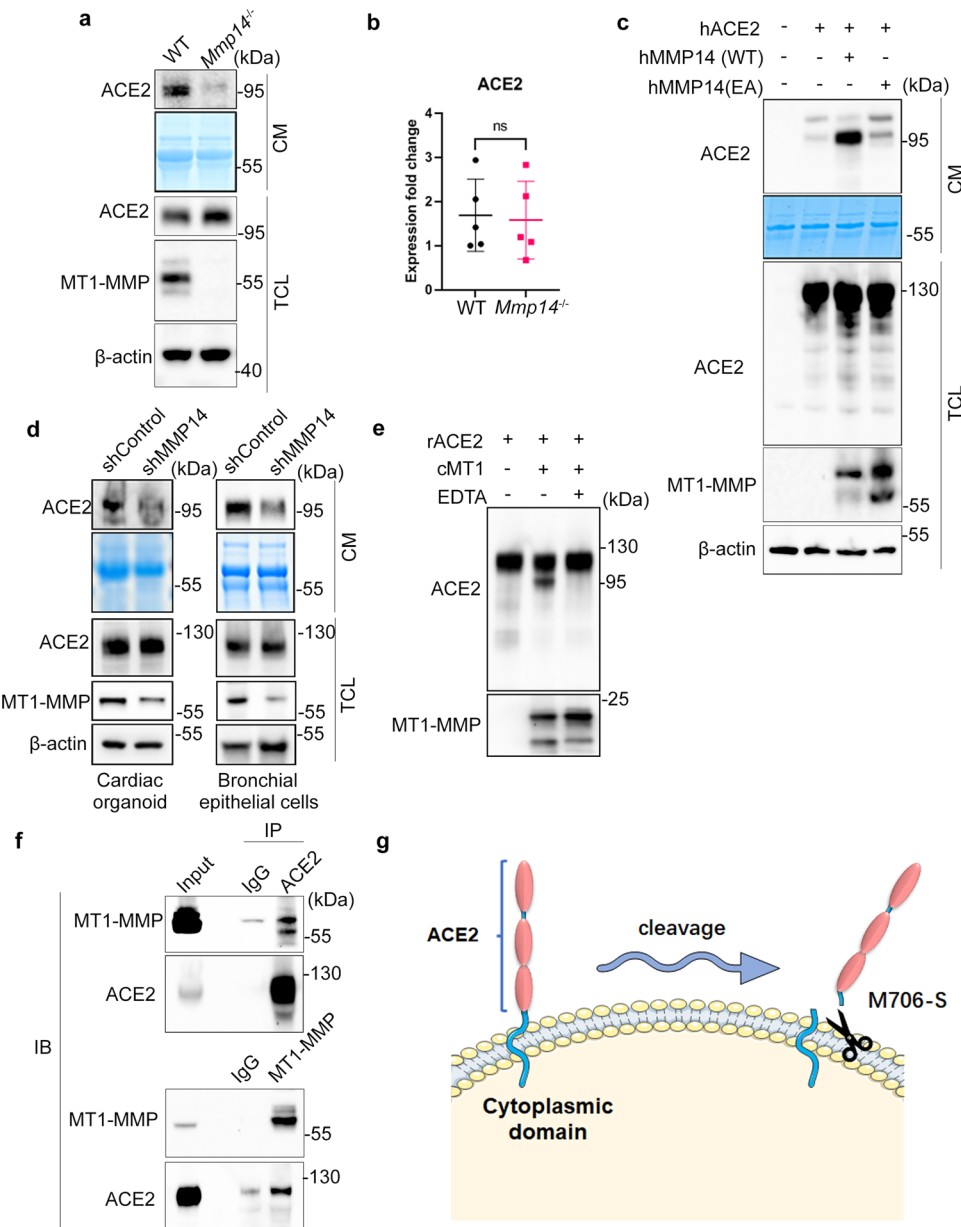

**Fig. 1 | The shedding of ACE2 by MT1-MMP. a** Western blotting analyses on the expression of ACE2 in the total cell lysate (TCL) and conditioned medium (CM) derived from wild-type and *Mmp14*$^{-/-}$ mouse primary lung epithelial cells. Coomassie stained membrane served as a loading control for conditioned medium (*n* = 3). **b** Real-time quantitative PCR analyses on the relative mRNA expression of ACE2 in wild-type and *Mmp14*$^{-/-}$ mouse primary lung epithelial cells (*n* = 5). **c** HEK293T cells were co-transfected with human ACE2 and wild-type or E/A catalytic mutant MT1-MMP (MT1 E240A). Expression of full-length ACE2 in total cell lysates and conditioned medium was detected by western blotting. Coomassie stained membrane was a loading control for conditioned medium (*n* = 3). **d** Western blotting analyses on the expression of solACE2 and memACE2 in human cardiac organoids and human bronchial epithelial cells that were lentivirally transduced with either shRNA or shMMP14 (*n* = 3). **e** The recombinant full-length ACE2 (rACE2) was incubated with the catalytic domain of MT1-MMP (cMT1) and the protein mixture was analyzed by western blotting using specific antibodies (*n* = 3). **f** Endogenous interaction between MT1-MMP and ACE2. Co-immunoprecipitation was performed with the cell lysates of Caco2 cells and examined by western blotting analyses using specific antibodies. IgG immunoprecipitates were used as controls (*n* = 3). **g** A diagram illustrating the cleavage of ACE2 by MT1-MMP. Data are means ± S.E.M. of three independent repeats; two-tailed unpaired *t*-test for (**b**). Source data are provided as a Source Data file.

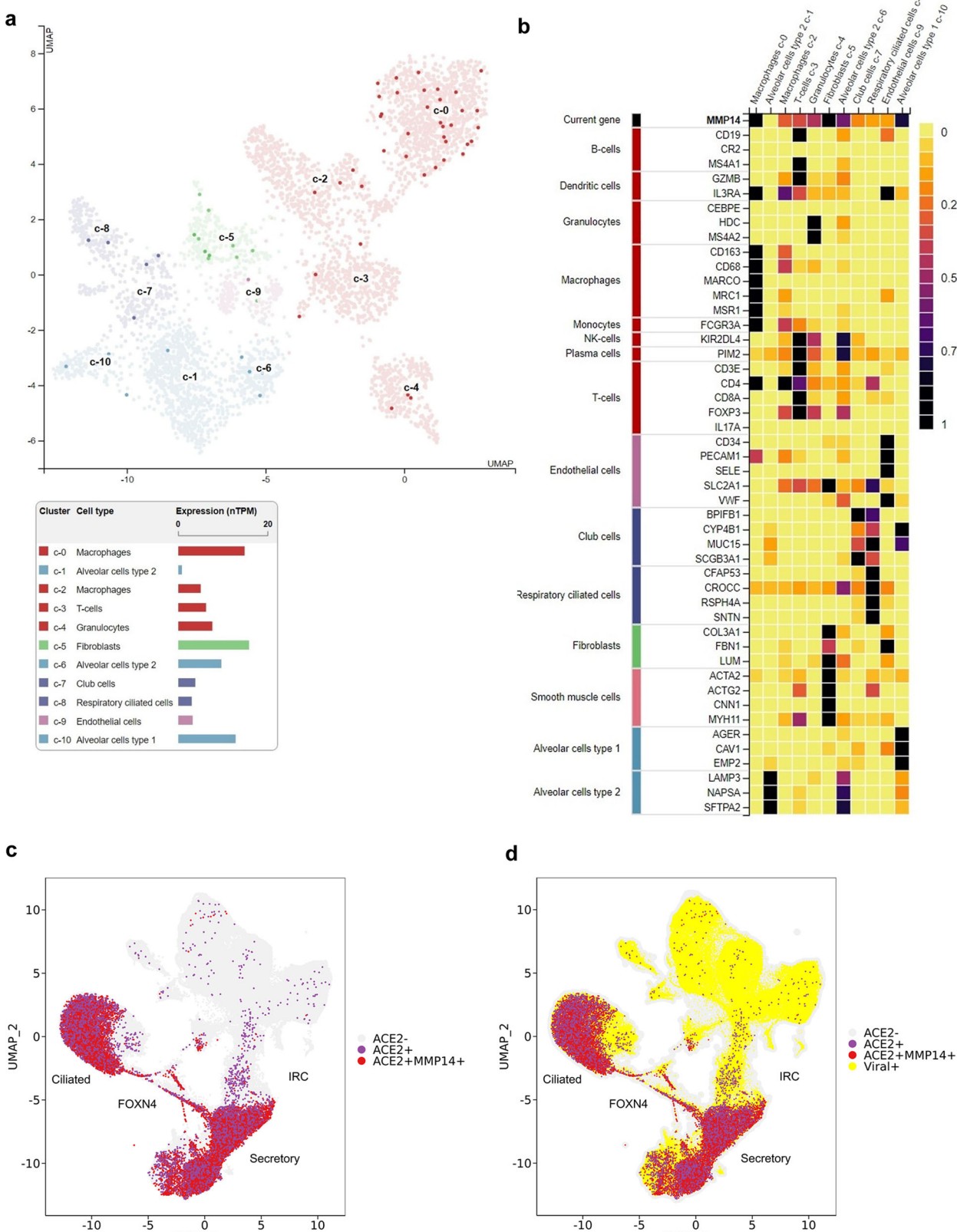

**Fig. 2 | MT1-MMP⁺ACE2⁺ lung epithelial cells are susceptible to SARS-CoV-2 infection. a** MT1-MMP⁺ cells were distributed across the indicated cell types in the human lung within UMAP. **b** Heat map analysis of *MMP14* gene expression in various lung and airway cells. **c** The distribution of ACE2⁺ and ACE2⁺ MT1-MMP⁺ cells in the lung was assessed using human cellular profiles at the single cell level. Gene expression was visualized for each cell type using tSNE. **d** UMAP is depicted for ACE2⁺ (purple), ACE2⁺MT1-MMP⁺ cells (red) and viral RNA⁺ cells (yellow circles).

indeed overlapping in human lung. To further investigate whether MT1-MMP⁺ ACE2⁺ cells are susceptible to SARS-CoV-2 infection, we compared the expression patterns between MT1-MMP and ACE2 in lungs from infected patients. MT1-MMP was colocalized with ACE2 in respiratory epithelial cells, such as ciliated and secretory cells, that were highly susceptible to the infection of SARS-CoV-2 (Fig. 2c, d).

As ACE2 is a well-known receptor for SARS-CoV-2 and we observed that MT1-MMP promotes the shedding of ACE2, we inevitably postulated that MT1-MMP may regulate the infection of SARS-CoV-2. By analyzing the single cell-transcriptome dataset, we found that the expression of *MMP14* was significantly elevated in lungs from patients infected with SARS-CoV-2 (Fig. 3a). To further investigate the interplay between ACE2 and MT1-MMP, we studied the role of MT1-MMP in ACE2-mediated viral infection using pseudotyped virus. ACE2-expressing HEK293T cells were transfected with either empty vector, wild-type or catalytically inactive MT1-MMP and subjected to infection of SARS-CoV-2 pseudotyped virus (Fig. 3b). We found that ectopic expression of wild-type MT1-MMP, but not the catalytically inactive MT1-MMP, promoted SARS-2-S-driven entry (Fig. 3b). Blockade of ACE2 shedding by the ablation of MT1-MMP cleavage site through S707A mutation abolished the promoting effect of MT1-MMP on the pseudotype entry whereas mutation at I727A that was irrelevant to MT1-MMP cleavage had negligible effects on the virus entry (Fig. 3b; Supplementary Fig. 3a), showing that MT1-MMP regulates SARS-CoV-2 entry by ACE2 shedding. To further confirm that solACE2 released by MT1-MMP facilitates pseudotype entry, Caco2 cells were inoculated with the conditioned media derived from 293T cells transfected with either empty vector, ACE-2 only or ACE2 plus MT1-MMP. We found that only the conditioned medium derived from 293T cells co-expressing ACE2 and MT1-MMP promoted SARS-2-S-driven entry in a concentration-dependent manner (Fig. 3c). Conversely, knockdown of MT1-MMP effectively suppressed the pseudotype entry in Huh7 and Caco2 cells (Fig. 3d, e). Infection of pseudotyped virus led to increased ACE2 shedding in Caco2 cells, which was completely suppressed by MT1-MMP depletion (Fig. 3f). Furthermore, we found that solACE2 released by MT1-MMP bound to SARS-CoV-2 spike proteins (Supplementary Fig. 6a). Previous studies showed that solACE2 facilitates the cell entry of SARS-CoV-2 through AT1/AVPR1B-mediated endocytosis. To investigate whether solACE2-MT1-MMP pathway adopts a similar strategy for promoting SARS-CoV-2 entry, we knocked down either AT1 or AVPR1B in Caco2 cells. We found that only knockdown of AT1 would attenuate the MT1-MMP-induced increase in pseudotype entry (Supplementary Fig. 6b), suggesting that solACE2-MT1-MMP pathway is predominantly mediated by AT1. To further confirm that SARS-CoV-2 may enter cells via AT1-mediated endocytosis, non-permissive 293T cells were firstly transfected with a plasmid encoding AT1, followed by the inoculation with recombinant spike (S) of SARS-CoV-2 and solACE2 from the conditioned media of 293T cells expressing ACE2 only or ACE2 with MT1-MMP. Immunofluorescent staining revealed that AT1 was mainly localized on the plasma membrane of the transfected 293T cells (Supplementary Fig. 6c). The addition of recombinant S coupled with the conditioned medium from 293T cells co-expressing ACE2 and MT1-MMP led to reduced membrane localization of AT1 and increased puncta-staining pattern in the cytosol (Supplementary Fig. 6c). Importantly, S and solACE2 were colocalized with AT1 at the puncta (Supplementary Fig. 6c), indicative of the internalization of solACE2/S/AT1 complex. In contrast, solACE2/S complex was not observed in non-permissive 293T cells inoculated with recombinant S and the conditioned medium of 293T cells expressing ACE2 only (Supplementary Fig. 6c). These results suggest that solACE2 released by MT1-MMP facilitates the internalization of SARS-CoV-2 Spike via AT1. To further examine whether solACE2 released by MT1-MMP forms a complex with S and AT1, AT1-expressing 293T cells were inoculated with the conditioned media derived from 293T cells co-expressing MT1-MMP and ACE2. Upon the addition of recombinant S, co-immunoprecipitation experiment was

performed. Both ACE2 and S could be detected in the AT1 immuno-precipitate (Supplementary Fig. 6d), suggesting the complex formation of sACE2/S protein/AT1.

We further validated our findings obtained from pseudotyped virus with authentic SARS-CoV-2 virus. Consistent with the data of pseudotyped virus, the conditioned medium derived from 293T cells co-expressing ACE2 and wild-type MT1-MMP promoted the entry of SARS-CoV-2 virus in HK-2 cells (Fig. 3g), a human renal tubule cell line with high susceptibility to SARS-CoV-2 infection, suggesting that solACE2 released by MT1-MMP indeed facilitates viral entry. Furthermore, the addition of recombinant ACE2 (rACE2) comprising human ACE2 with the deletion of transmembrane and cytosolic domains (ACE2$_{1-740}$) at the physiological dosage enhanced SARS-CoV-2 infectivity in HK-2 cells; this stimulating effect was further exacerbated with rACE2 digested by catalytic domain of MT1-MMP (Fig. 3h). On the contrary, inhibition of MT1-MMP activity by shRNA-mediated knockdown effectively attenuated the entry of SARS-CoV-2 in human bronchial epithelial cells (Fig. 3i; Supplementary Fig. 7a). These results indicate that solACE2 released by MT1-MMP mediates SARS-CoV-2 entry in vitro. To investigate whether solACE2-mediated entry of SARS-CoV-2 depends on memACE2, we synthesized recombinant ACE2 comprising M1-M706 (rACE2$_{1-706}$) that corresponded to solACE2 released from MT1-MMP cleavage site. A549 and HEK293T cells, two well-known SARS-CoV-2 nonpermissive cell lines due to lack of ACE2 expression, were inoculated with rACE2$_{1-706}$ and then subjected to SARS-CoV-2 infection. Without the presence of rACE2$_{1-706}$, neither viral antigens nor gene copies were detected in both cell lines (Fig. 4a–c). The addition of rACE2$_{1-706}$ in a dose-dependent manner promoted the infection of SARS-CoV-2 in both cells (Fig. 4a–c). These results suggest that solACE2 released by MT1-MMP facilitates SARS-CoV-2 infection in a memACE2-independent manner.

To investigate whether solACE2 released by MT1-MMP facilitates SARS-CoV-2 entry in vivo, we designed an adeno-associated viral vector under the control of CMV promoter, a potent promoter commonly used for mediating the expression of transgenes, to ectopically and systemically express either soluble form of human ACE2 comprising M1-M706 (solACE2$_{1-706}$) that corresponded to solACE2 released from MT1-MMP cleavage site or human ACE2 with the deletion of transmembrane and cytosolic domains (solACE2$_{1-740}$) or full length membrane form of ACE2 (ACE2$_{FL}$) as a positive control or blank control in wild-type C57BL/6 mice that were naturally resistant to infection with wild-type SARS-CoV-2. SARS-CoV-2 cannot infect laboratory mice as a result of the incompatibility between SARS-CoV-2 spike proteins and mouse ACE2. Mice were firstly transduced with adenoviral particles for 14 days for facilitating stable expression of different forms of solACE2, followed by challenge with wild-type SARS-CoV-2 for another 2 days (Fig. 4d). Upon adenoviral transduction, human ACE2 mRNA could be detected in different organs with the highest level in liver among the examined tissues (Supplementary Fig. 8a). ELISA assay using specific antibodies against human ACE2 revealed that the levels of human solACE2 in plasma from solACE2$_{1-706}$-expressing mice and solACE2$_{1-740}$-expressing mice were 9.85 ng/ml and 17.33 ng/ml respectively (Supplementary Fig. 8b), both of which were within the physiological range of serum ACE2 levels in human (3–40 ng/ml). To investigate whether solACE2 facilitates SARS-CoV-2 entry, we detected the expression of human ACE2 and SARS-CoV-2 Nucleocapsid Protein (NP) by immunofluorescent staining (Fig. 4e). Unlike control mice, which were completely insusceptible to SARS-CoV-2 infection even with high infectious doses, viral antigens could be detected in lung sections of mice expressing solACE2$_{1-706}$/ solACE2$_{1-740}$ / ACE2$_{FL}$ upon intranasal infection of SARS-CoV-2 (Fig. 4e). Notably, SARS-CoV-2 NP was highly colocalized with human ACE2 in alveolar linings of mice expressing solACE2$_{1-706}$/ solACE2$_{1-740}$ (Fig. 4e), indicative of the internalization of human solACE2/SARS-CoV-2 viral complex in alveolar cells. Importantly, the level of this viral complex in the lungs of

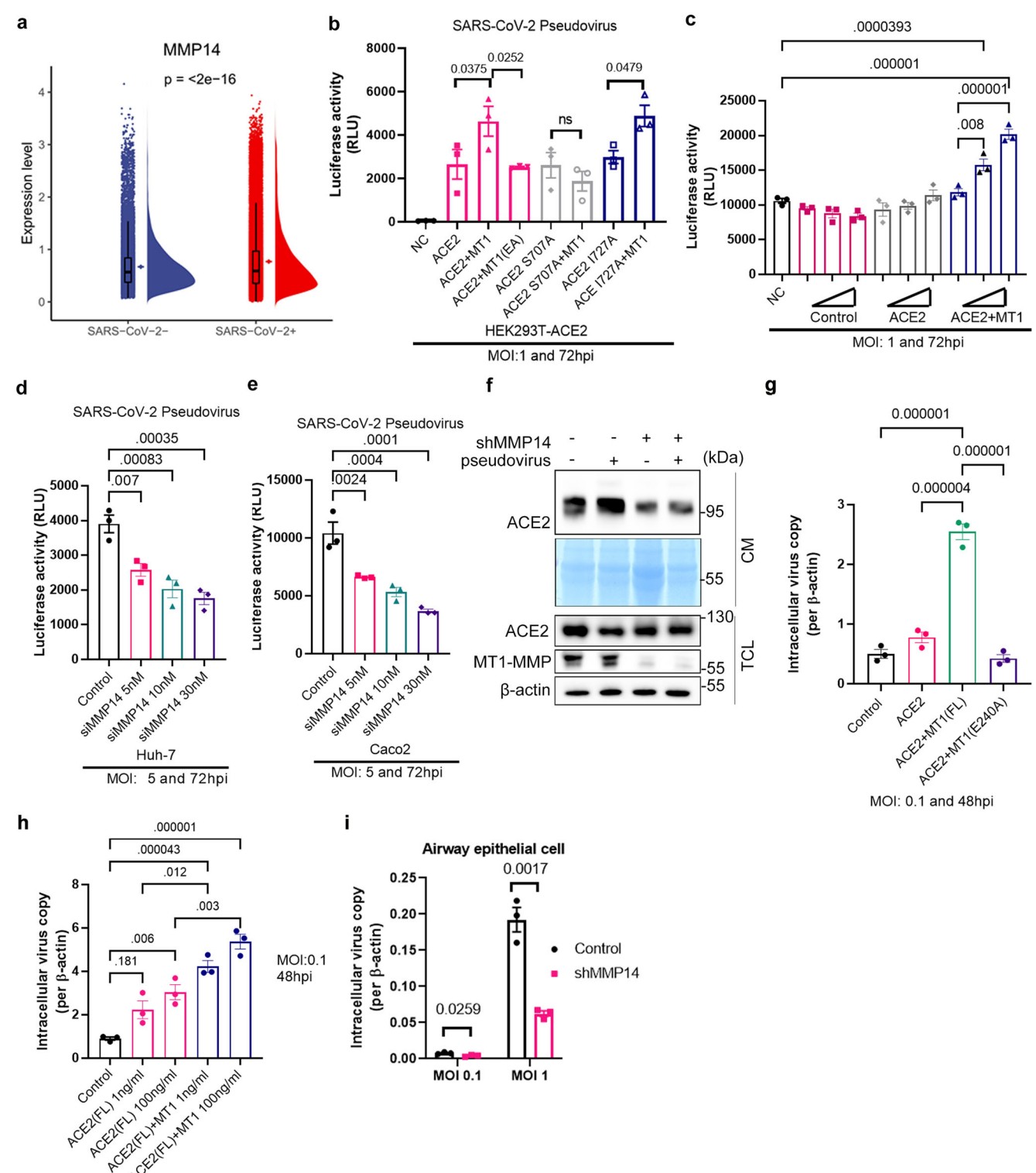

solACE2_{1-706}-expressing mice was considerably higher than that in solACE2_{1-740}-expressing mice (Fig. 4e; supplementary Fig. 9a). Concordant with the antigen detection, viral gene copies were significantly elevated in lungs of solACE2_{1-706}-expressing mice when compared to solACE2_{1-740}-expressing mice (Fig. 4f), suggesting that physiologically relevant solACE2_{1-706} is more efficient than solACE2_{1-740} in mediating virus entry. In addition, solACE2_{1-706}-expressing mice exhibited the similar levels of viral loads (Fig. 4e, f), tissue damages (e.g. perivascular to interstitial inflammatory cell infiltrates, necrotic cell debris, and alveolar edema) (Fig. 4g, h; supplementary Fig. 9a) and lung

inflammatory responses (supplementary Fig. 9b) as those observed in ACE2 _{FL}-expressing mice at 2 dpi. These data suggest that solACE2 released by MT1-MMP is as important as full-length memACE2 in mediating SARS-CoV-2 infection in vivo. The tissue damages resulted from viral infection in our experimental model reveal that solACE2-directed infections are physiologically relevant.

To explore the therapeutic potential of targeting MT1-MMP in the treatment of SARS-CoV-2 infection, we examined the effect of MT1-MMP inhibition on SARS-CoV-2 infection in human organoid culture systems that provided a physiologically relevant platform for SARS-

**Fig. 3 | MT1-MMP regulates the cell entry of SARS-CoV-2 by releasing solACE2.** **a** Comparison of *MMP14* expression levels in lung and airway cells from uninfected populations and patients infected with SARS-CoV-2. Boxplots show the median (central line), the 25–75% interquartile range (IQR) (box limits), the ±1.5 × IQR (whiskers). (*n* = 44 for healthy controls; *n* = 104 for infected patients) **b** MT1-MMP promotes the entry of SARS-CoV-2 viral pseudotypes in host cells. HEK293T cells expressing different ACE2 mutants (wild-type ACE2, ACE2 S707A & ACE2 I727A) were transfected with WT or E/A catalytic mutant MT1-MMP (MT1 E240A) and infected with SARS-CoV-2 viral pseudotypes for 72 h. Intracellular luciferase levels were measured. (*n* = 3) **c** ACE2-HEK293T cells were inoculated with various concentrations of conditioned media derived from HEK293T transfected with ACE2 with/without MT1-MMP, followed by infection with SARS-CoV-2 pseudotypes for 72 h. Intracellular luciferase levels were measured. (*n* = 3) **d, e** Huh-7 or Caco2 cells were transfected with either siRNA targeting MMP14 or control siRNA, followed by the infection with SARS-CoV-2 pseudotyped virus. Luciferase levels were detected by chemiluminescence at 72 h post-infection. (*n* = 3) (p=<2e-16) **f** Caco2 cells stably transduced with shMMP14 or control shRNA were infected with SARS-CoV-2 pseudotyped virus for 48 h. The expression levels of solACE2, memACE2 and MT1-MMP were subsequently detected by western blotting. (*n* = 3) **g** The HK-2 cells were inoculated with the conditioned media derived from HEK293T cells co-expressing either ACE2, ACE2 with MT1-MMP or ACE2 with catalytic mutant MT1-MMP (E240A), followed by SARS-CoV-2 infection. (*n* = 3) **h** The HK-2 cells pretreated with different concentrations of recombinant ACE2 and with or without incubation with catalytic domain of MT1-MMP were infected with SARS-CoV-2. Mock-treated HK-2 cells served as a control. (*n* = 3) **i** Primary human bronchial epithelial 3D cultures with an air-liquid interface (ALI) were lentivirally transduced with shRNA targeting MT1-MMP or control shRNA, following by the infection of SARS-CoV-2. (*n* = 3) The data are shown as the means ± SEM from three independent experiments. *P* values were calculated using one-way ANOVA for (**b**, **e**; **g**, **h**); two-way ANOVA for (**i**) and unpaired two -tailed *t*-test for (**a**). Source data are provided as a Source Data file.

CoV-2 translational research[39–41]. We found that knockdown of MT1-MMP markedly suppressed the entry of SARS-CoV-2 in multiple human organoid models including primary human bronchial epithelial 3D cultures with an air-liquid interface (ALI), human cardiac organoids formed by iPSC-derived cardiomyocytes and human liver organoids (Fig. 5a–c; Supplementary Fig. 7b, c). Similarly, inhibition of MT1-MMP activity by 3A2, a well-characterized antibody against MT1-MMP with remarkable neutralizing abilities[42–44], inhibited solACE2 release and SARS-CoV-2 entry in ALI cultures in a dose-dependent manner (Fig. 5d; Supplementary Fig. 10a). To investigate whether MT1-MMP similarly controls the cell entry of SARS-CoV-2 variants of concern (VoC), human bronchial epithelial cells with MT1-MMP knockdown were inoculated with the omicron SARS-CoV-2 variant, a dominant variant circulating globally. Depletion of MT1-MMP significantly reduced the infectivity of omicron variant in human bronchial epithelial cells (Fig. 5e). Consistently, inhibition of MT1-MMP activity by siRNA-mediated knockdown similarly abolished the cell entry of SARS-CoV-2 wild-type virus and its VoCs including Delta and Omicron in Caco2 cells (Supplementary Fig. 10b, d). These results collectively showed that MT1-MMP is a major host factor controlling the cell entry of SARS-CoV-2, highlighting MT1-MMP as a potential druggable target for the treatment of COVID-19 and its comorbidities.

As ageing is a primary risk factor of SARS-CoV-2 infection[45,46] and membrane-bound MT1-MMP is found to be shed from the cell surface to produce a functional soluble form in both physiological and pathological events[47–49], we therefore examined the concentrations of solACE2 and circulating MT1-MMP in plasma derived from young adults with average age of 29.2 years and the elderly with average age of 80.1 years. We found that both solACE2 and circulating MT1-MMP are markedly increased in plasma of aged human subjects (Fig. 6a, b). In line with elevated level of circulating MT1-MMP, analyses of single cell-transcriptome data revealed that the expression of membrane-bound *MMP14* was upregulated in the lungs of the elderly (Fig. 6c). Moreover, there was a significantly positive correlation between plasma MT1-MMP and solACE2 (Fig. 6d). Similar observations in the upregulation of ACE2 and MT1-MMP were found in aged mice (Supplementary Fig. 11a, b). As there is no *ACE2* mRNA expression differences correlating with age[50], the elevated level of solACE2 in the elderly is likely attributed to increased shedding of ACE2 probably mediated by MT1-MMP. These data reinforce the role of MT1-MMP in ACE2 shedding in human.

To investigate whether the in vitro antiviral effect of MT1-MMP inhibition observed in primary human cells and organoids translates to in vivo efficacy against SARS-CoV-2 infection in the ageing situation, aged wild-type mice at 18 months of age were prophylactically administrated with 3A2 and intranasally challenged with the mouse-adapted strain of SARS-CoV-2 derived from SARS-CoV-2 N501Y-carrying variants (Fig. 7a). Infected mice were therapeutically treated with 3A2 at 1dpi and sacrificed for further analyses at 2 dpi. We have previously shown that 3A2 treatment confers robust protection against diet-induced obesity and age-associated insulin resistance in mice via inhibition of MT1-MMP activities without the induction of adverse effects[25,51], demonstrating the in vivo efficacy of this antibody in targeting MT1-MMP. As expected, inhibition of MT1-MMP via 3A2 treatment nearly depleted solACE2 in the plasma from aged mice (Fig. 7b). In mice treated with IgG controls, analyses of viral gene copies and antigens revealed that high viral loads were detected in lungs and tracheas at 2 dpi (Fig. 7c, d; Supplementary Fig. 12a). In alignment with reduced solACE2 levels, viral loads were dramatically reduced in both lungs and tracheas of mice treated with 3A2 (Fig. 7c, d; Supplementary Fig. 12a). Under the infection with high viral titers, the control group gradually lost weight and some of them died starting on 1dpi, but all mice treated with 3A2 survived and exhibited reduced loss of body-weight (Fig. 7e, f). Histopathological analyses showed epithelial damage, vascular congestion as well as submucosal and bronchiolar infiltration in vehicle-treated mice (Fig. 7g, h). MT1-MMP inhibition remarkably alleviated these pathological changes in lungs of infected mice (Fig. 7g, h). Furthermore, 3A2 reduced the production of inflammatory cytokines and chemokines in the lung tissues (Supplementary Fig. 12b). These results show that pharmacological inhibition of MT1-MMP can effectively protect aged mice from SARS-CoV-2 infection and tissue damages.

## Discussion

With the tremendous economic and healthcare burdens caused by the outbreak of COVID-19 and the waning of vaccine protection against SARS-CoV-2 variants, there is an urgent need to identify specific and druggable targets for treating this disease. It is therefore imperative to enhance our understanding of the cell entry mechanism of SARS-CoV-2 in order to inform intervention strategies. While increasing evidence links the level of solACE2 with the infectivity of SARS-CoV-2, the host factors and the underlying mechanisms regulating the release of ACE2 from the cell surface remain elusive. We herein identified a previously unappreciated mechanism for the cell entry of SARS-CoV-2, involving ectodomain shedding of ACE2 by MT1-MMP. We showed that MT1-MMP is a host factor mediating the generation of solACE2 and increased shedding of ACE2 majorly mediated by MT1-MMP contributes to the cell entry of SARS-CoV-2.

Currently, little is known about the function of MT1-MMP in SARS-CoV-2 infection. We showed herein that SARS-CoV-2 infection leads to the increased activation of MT1-MMP that directly cleaves membrane ACE2 at M706-S to release soluble ACE2. After the removal of signal peptide, solACE2 released by MT1-MMP should comprise amino acid sequence of Q18-M706. MT1-MMP promotes the cell entry of SARS-CoV-2 by the release of solACE2 as blockade of ACE2 shedding by the ablation of MT1-MMP cleavage site abrogates the stimulating effect of MT1-MMP on the cell entry of SARS-CoV-2. Previous studies reveal that the solACE2 facilitates the infection by receptor-mediated

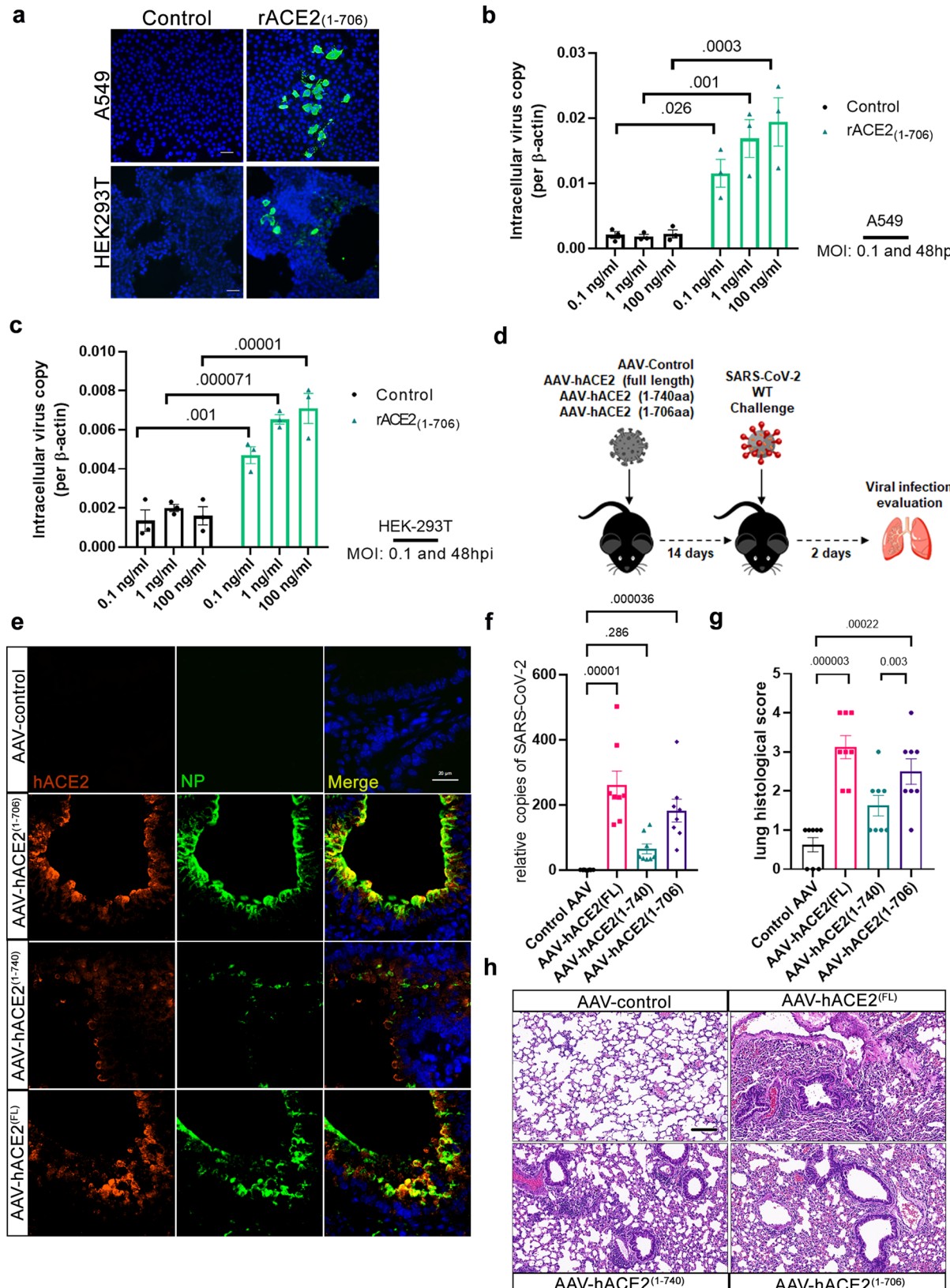

endocytosis[16]. In our current study, we further demonstrated that solACE2 released by MT1-MMP binds to Spike proteins of SARS-CoV-2, which facilitates the cell entry of SARS-CoV-2 through AT1-mediated endocytosis. Despite the respiratory route being dominant in the infection of SARS-CoV-2, only a small population of the lung cells expresses detectable level of ACE2. However, SARS-CoV-2 has

organotropism beyond the respiratory tract, evidenced by the facts that SARS-CoV-2 infection leads to multiorgan injury and high levels of ACE2 can be detected in kidney, heart muscle and small intestine[52]. We therefore postulate that SARS-CoV-2 adopts MT1-MMP/solACE2-mediated machinery to infect cells with the relatively low expression of ACE2 or without ACE2 expression, facilitating multiorgan infections.

**Fig. 4 | MT1-MMP-mediated ACE2 shedding mediates the SARS-CoV-2 cell entry in vitro and in vivo. a–c** The A549 and HEK293T cells were inoculated with rACE2$_{1-706}$, followed by SARS-CoV-2 infection. Vehicle-treated cells served as a control. The viral antigens of these cells were detected by immunostaining of Nucleocapsid Protein (NP) (green) as shown in **a**. (One representative data was shown from three independently repeated experiments) (Scale bar: 20 μm) The viral gene copies of these cells were measured by qPCR analyses of the SARS-CoV-2 genes as shown in **b** and **c**. ($n = 3$ biologically independent experiments) **d** Experimental strategy to study the role of human solACE in mediating SARS-CoV-2 infection in mice. **e** Immunostaining of human ACE2 (red) and SARS-CoV-2 (green) Nucleocapsid Protein (NP) in lung sections of mice expressing solACE2$_{1-706}$/ solACE2$_{1-740}$/ACE2$_{full length (FL)}$ ($n = 8$ for AAV-control; $n = 8$ for AAV-hACE2(1-706); $n = 8$ for AAV-hACE2 (1-740); $n = 8$ for AAV-hACE2(full length)).

(Scale bar: 20 μm) **f** The viral loads in the lungs of mice expressing solACE2$_{1-706}$/ solACE2$_{1-740}$/ ACE2$_{full length (FL)}$ were measured by qPCR analyses of the SARS-CoV-2 genes ($n = 8$ for AAV-control; $n = 8$ for AAV-hACE2(1-706); $n = 8$ for AAV-hACE2 (1-740); $n = 8$ for AAV-hACE2(full length)). **g** Summary histology scores determined in the lung tissues of mice expressing solACE2$_{1-706}$/ solACE2$_{1-740}$/ACE2$_{full length (FL)}$ ($n = 8$ for AAV-control; $n = 8$ for AAV-hACE2(1-706); $n = 8$ for AAV-hACE2 (1-740); $n = 8$ for AAV-hACE2(full length)) **h** Representative Hematoxylin-eosin (HE) staining of lungs of mice expressing solACE2$_{1-706}$/ solACE2$_{1-740}$/ACE2$_{full length (FL)}$ ($n = 8$ for AAV-control; $n = 8$ for AAV-hACE2(1-706); $n = 8$ for AAV-hACE2 (1-740); $n = 8$ for AAV-hACE2(full length)). (Scale bar: 50 μm) Data are means ± S.E.M. of three independent repeats. Statistical analyses were performed by one way ANOVA for (**b, c; f, g**) Source data are provided as a Source Data file.

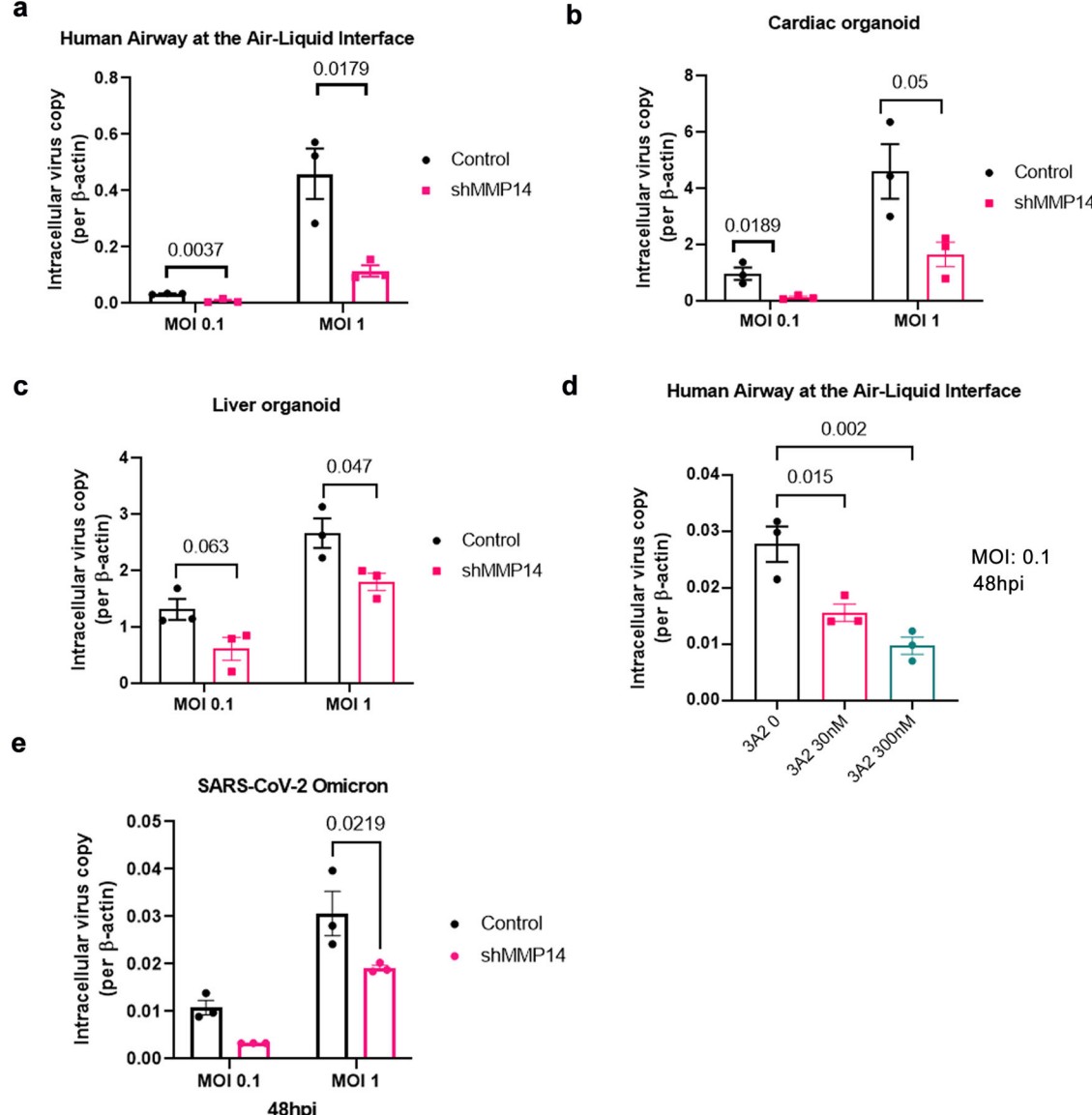

**Fig. 5 | MT1-MMP inhibition suppresses SARS-CoV-2 infection in human organoids. a–e** The inhibitory effect of MT1-MMP knockdown on SARS-CoV-2 infectivity in primary cells and organoids. Primary human bronchial epithelial 3D cultures with an air-liquid interface (ALI) ($n = 3$ biologically independent experiments), human cardiac organoids formed by iPSC-derived cardiomyocytes ($n = 3$) and human liver organoids ($n = 3$) were lentivirally transduced with shRNA targeting MT1-MMP or control shRNA, following by the infection of SARS-CoV-2. **d** Air–liquid interface

culture was pretreated with increasing dosages of 3A2 and infected with SARS-CoV-2. ($n = 3$ biologically independent experiments) **e** Human bronchial epithelial cells transduced with shMT1-MMP or shRNA control were infected with SARS-CoV-2 omicron variant. ($n = 3$ biologically independent experiments) Data are means ± S.E.M. of three independent repeats. Statistical analyses were performed by one way ANOVA for (**d**) and multiple unpaired two-tailed $t$-test for (**a–c**), **e**. Source data are provided as a Source Data file.

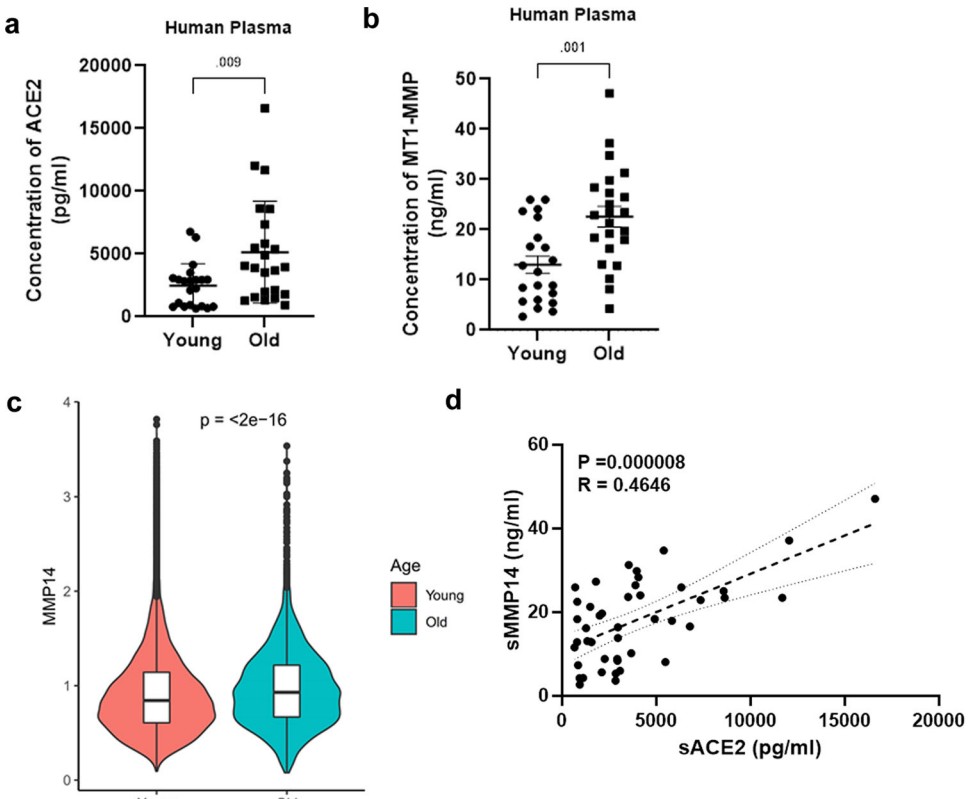

**Fig. 6 | The positive correlation between MT1-MMP and ACE2 in the elderly. a, b**
Expression levels of soluble ACE2 and MT1-MMP in the plasma of young human subjects and aged human subjects were measured by ELISA ($n = 21$ young people & 23 old people) (unpaired two-tailed $t$-test). The data are shown as the means ± SEM. **c** Comparison of MMP14 expression levels in young human subjects and elderly human subjects by analysis of single-cell transcriptome obtained from human lung and airway cells (unpaired two-tailed $t$-test). Boxplots show the median (central

line), the 25–75% interquartile range (IQR) (box limits), the ±1.5 × IQR (whiskers). ($n = 25$ for young subjects; $n = 11$ for old subjects) **d** The positive correlation between plasma MT1-MMP and soluble ACE2 in human plasma ($p = 000008$, two-tailed; $R = 0.4646$, $n = 43$; Spearman's correlation) Dashed lines represent the 95% confidence interval for the lines of regression presented in the figure. Source data are provided as a Source Data file.

Indeed, our studies have shown that solACE2 released by MT1-MMP facilitates the cell entry of SARS-CoV-2 in non-permissive cells that express undetectable level of ACE2. More importantly, ectopic expression of human solACE2 enabled SARS-CoV-2 infection in naturally insusceptible mice. Moreover, blockade of ACE2 shedding by MT1-MMP inhibition conferred strong therapeutic effects against SARS-CoV-2 infection in aged mice, confirming the essentiality of solACE2 for SARS-CoV-2 infection in vivo. These findings confirm MT1-MMP as a major sheddase of ACE2 and provide the in vivo evidence showing MT1-MMP-mediated release of solACE2 as an alternative entry pathway for SARS-CoV-2.

The level of circulating ACE2 correlates with the severity of COVID-19 and predicts mortality[53–55]. Modulating the release of solACE2 may therefore provide insights into the treatment strategies for COVID-19. In our study, we also reported that solACE2 level in the plasma was elevated significantly in the elderly, the population with the highest risk of severe illness with COVID-19 (Fig. 5). We further showed the positive correlation between solACE level and circulating MT1-MMP, suggesting that the elevated level of solACE2 observed in the elderly and the patients with COVID-19 is likely attributed to increased shedding of ACE2 predominantly mediated by MT1-MMP. Inhibition of MT1-MMP effectively reduced the release of solACE2 and suppressed the infection of SARS-CoV-2 wild-type strain and its variants of concern in human primary cells and organoids. Moreover, MT1-MMP blockade conferred protection against SARS-CoV-2 infection in aged mice. Given the strong inhibitory effect conferred by MT1-MMP inhibition on the cell entry of SARS-CoV-2, MT1-MMP can serve as a potential therapeutic target for the treatment of COVID-19 by

controlling the release of solACE2. ACE2 is widely expressed in different tissues and involved in various biological proesses, such as the control of blood pressure by angiotensin[56]. Therefore, the regulation of ACE2 by MT1-MMP may be physiologically relevant to other biological functions in addtion to SARS-CoV-2 cell entry, which deserves further investigation in the future.

We demonstrate that ACE2 is a direct substrate of MT1-MMP, and the MT1-MMP cleavage of ACE2 regulates cell entry of SARS-CoV-2 in vitro and in vivo. ACE2 cleavage has been identified for years. However, the major protease responsible for ACE2 cleavage remains to be elucidated. Inhibition of ADAM17, a transmembrane metalloprotease reported to cleave ACE2, only minorly reduced solACE2 levels[17,57,58]. Furthermore, the role of ADAM17 in ACE2 shedding has not been validated in vivo[17,57,58]. In fact, we showed that inhibition of MT1-MMP leads to dramatic reduction in solACE2 generation in vitro and in vivo accompanied by robust suppression of SARS-CoV-2 infection. Furthermore, we found that MT1-MMP was much more efficient than ADAM17 or other ADAM family members in releasing solACE2 in cells (Supplementary Fig. 13). These findings uncover MT1-MMP as a primary sheddase of ACE2 and the cleavage of ACE2 by MT1-MMP is physiologically relevant with immediate clinical implications. We previously demonstrated that the interplay between MT1-MMP and ADAM family members, such as ADAM9 and ADAM15, plays a significant role in the regulation of various biological processes, such as calvarial development and angiogenesis[27,59]. It is therefore possible that MT1-MMP and ADAM17 may work synergistically in the regulation of ACE2 shedding and hence the infection of SARS-CoV-2.

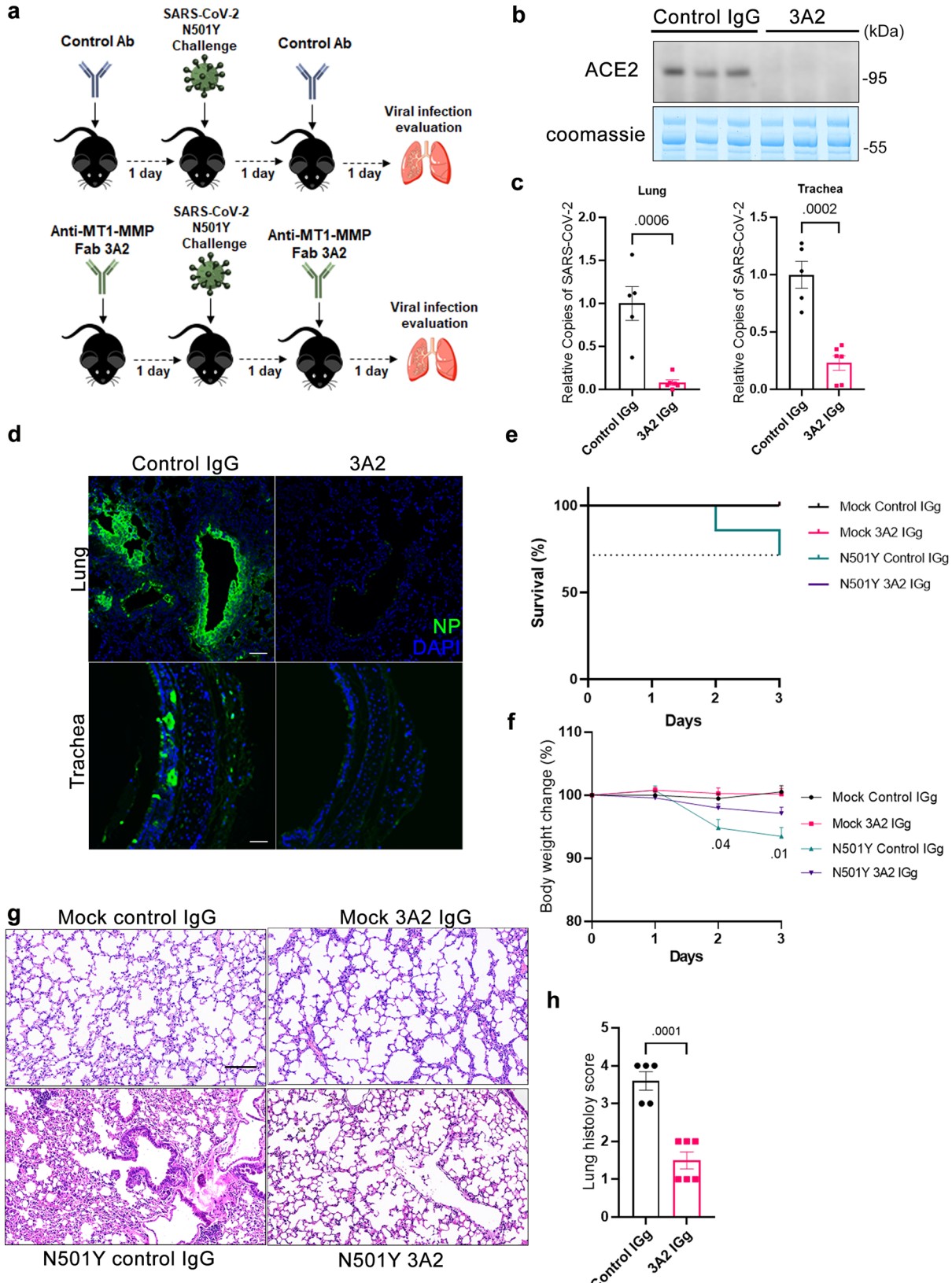

The entry of SARS-CoV-2 into the host cells is facilitated by the priming of SARS-CoV-2 S proteins and the ectodomain shedding of ACE2[16,60]. These two proteolytic events essential for SARS-CoV-2 spread and infection are largely mediated by host proteases, raising the possibility that MT1-MMP may regulate SARS-CoV-2 entry through proteolytic processing of SARS-2-S in addition to its role in ACE2 shedding. To address this possibility, we examined the effect of MT1-MMP on the priming of SARS-CoV-2 S proteins. We found that ectopic expression of MT1-MMP does not alter the activation of S proteins from SARS-CoV-2 and its variants of concern (Supplementary Fig. 14a). Similarly, the catalytic domain of MT1-MMP could not efficiently cleave

**Fig. 7 | Pharmacological inhibition of MT1-MMP confers protection against SARS-CoV-2 infection in aged mice. a** Experimental strategy to study the beneficial effect of MT1-MMP inhibition on SARS-CoV-2 infection in aged mice. **b** The level of plasma ACE2 in 3A2/control IgG-treated mice challenged with mouse-adapted SARS-CoV-2 was detected by western blotting ($n = 3$ for IgG control; $n = 3$ for 3A2). **c** The viral loads in the lungs (left) and tracheas (right) from 3A2/control IgG-treated mice challenged with mouse-adapted SARS-CoV-2 were quantified by qPCR analyses of the SARS-CoV-2 viral genes ($n = 5$ for IgG control; $n = 6$ for 3A2). **d** Immunostaining of SARS-CoV-2 Nucleocapsid Protein (NP) (green) in lung sections of 3A2/control IgG-treated mice challenged with mouse-adapted SARS-CoV-2.

($n = 5$ for IgG control; $n = 6$ for 3A2) Scale bar: 20 μm. **e, f** Survival rate and weight changes in 3A2/control IgG-treated mice challenged with mouse-adapted SARS-CoV-2 ($n = 5$ for IgG control; $n = 6$ for 3A2). **g** Histological examination of lung sections from 3A2/control IgG-treated mice challenged with mouse-adapted SARS-CoV-2 by hematoxylin and eosin staining ($n = 5$ for IgG control; $n = 6$ for 3A2) (Scale bar: 20 μm). **h** Summary histology scores determined in the lung tissues from 3A2/control IgG-treated mice challenged with mouse-adapted SARS-CoV-2 ($n = 5$ for IgG control; $n = 6$ for 3A2). Data are means ± S.E.M. of three independent repeats. Statistical analyses were performed by unpaired two-tailed $t$-test for (**c, h**); one way ANOVA for (**f**). Source data are provided as a Source Data file.

recombinant S2 proteins derived from SARS-CoV-2 and VOCs (Supplementary Fig. 14b), suggesting that MT1-MMP-dependent control of viral entry is mainly mediated through ACE2 shedding. Even though MMP cleavage-fusion activation of spike proteins has been demonstrated to promote SARS-CoV-2 entry[61,62], our study suggests that other MMPs, but not MT1-MMP, may be involved in this cleavage process. The differential roles of different MMPs in SARS-CoV-2 infection certainly warrant further investigation in the future.

Emerging SARS-CoV-2 variants including Delta and Omicron have spread around the world. These variants have been confirmed to be much more infectious than the wild-type strain. Most of the countries with higher vaccination rates have been impacted by the Omicron variant of SARS-CoV-2. The high transmission rate of the Omicron variant is largely due to its ability to evade SARS-CoV-2 immunity induced by either vaccination or past infection[63–66]. Our study reveals that MT1-MMP could be the potential druggable target for these new variants as inhibition of MT1-MMP activity by either pharmacological antagonism or genetic knockdown effectively restricted the cell entry of SARS-CoV-2 wild-type virus and its variants of concern in human primary cells and organoids. Identification of new proteases involved in ACE2 processing will certainly provide potential opportunities to target proteases as an antiviral strategy.

COVID-19 is featured by an increased number of cases and a greater risk of severe disease with increasing age. The mechanism underlying the age-associated susceptibility to infection by SARS-CoV-2 remains largely unclear. Increased solACE2 levels observed in the aged population is considered as a risk factor for the age-associated susceptibility. Our findings do not only demonstrate MT1-MMP as a major sheddase of ACE2, but also show a significantly positive correlation between soluble MT1-MMP and solACE2 in both aged primates and non-primates. The robust inhibitory effect conferred by MT1-MMP inhibition on SARS-CoV-2 infection in aged mice reveals the potential contribution of MT1-MMP as a major host factor to the age-associated susceptibility to infection.

## Methods
### Mice
Wild-type mice on C57BL/6J background were from the Laboratory Animal Services Centre of The Chinese University of Hong Kong. *Mmp14*[-/-] mice in C57BL6 background kindly provided by Prof. Zhou Zhongjun have been described in the previous study[23]. All animals and their borne pups were housed in the animal house at Hong Kong Baptist University and maintained on a 12 h light/dark cycle with constant ambient temperature (22–24 °C) and humidity (~60%). They were fed with standard laboratory chow, and applied with water ad libitum. All animals were between 6 and 12 weeks old unless otherwise stated in the age group experiments. Mice of both sexes were used in the experiments. Primary cells and plasma were collected from mice at 13-day-old. All animal experiments were performed according to the guideline of the Committee on the Use of Human & Animal Subjects in Teaching & Research at Hong Kong Baptist University and procedures were approved by the Department of Health under Hong Kong legislation.

### Cell culture and transfection
Human embryonic kidney (HEK)293T, Caco-2, HK-2 cells, A549 and Huh-7 cells were purchased from the American Type Culture Collection (ATCC, USA; CRL-3216 for HEK293T; CRL-8024 for Huh7; HTB-37 for Caco2, CCL158 for A529). Human bronchial epithelial cells were purchased from ScienCell (cat no. 3210). HEK293T, Huh-7, and Caco2 cells were maintained in Dulbecco's modified Eagle's medium (DMEM) All culture media contained 10% fetal bovine serum (FBS, Gibco, USA) and were supplemented with 1% penicillin and streptomycin (Gibco, USA).

pCMV-GFP-ACE2 (HG10108-ACG) and pCMV-FLAG-MMP14 (HG10741-CF) plasmids were purchased from Sino Biological (CHINA). pCMV-FLAG-MMP14 (E240A), the primers used for plasmid construction were as follows: F: GGGGTACCATGTCTCCCGCCCCAAG; R: TCTA GACTCGAGTTTACTTATCGT; F1: TGGTGGCTGTGCACGCGCTGGGC CATGCCCT; R1: AGGGCATGGCCCAGCGCGTGCACAGCCACCA. PCR amplification was performed using the pCMV-FLAG-MMP14 plasmid as a template. The sequences of the gene-specific primers for constructing ACE2 point mutations were shown in Supplementary Table 1. All constructs were confirmed by sequencing. According to the manufacturer's instructions, the plasmids were transiently transfected into the indicated cells using Lipofectamine 3000 transfection reagents (Invitrogen, USA).

siRNA against MMP14 was purchased from Dharmacon (L-004145-00-0005, USA). According to the manufacturer's protocol, the siRNA was transfected into target cells with Lipofectamine RNAiMAX (Thermo Fisher, USA).

### Antibodies
The antibodies used in this study include the following: anti-SARS-CoV-2 spike antibody (40150-R007, Sino Biological, 1:1000); anti-ACE2 antibody (AF933, R&D System, 1:2000); anti-ACE2 antibody (EPR24705-45, ab272500, Abcam,1:500); anti-MMP14 antibody (EP1264Y, ab51074, Abcam, 1:2000); anti-FLAG tag antibody (66008-3-Ig, clone no. 2B3C4, Proteintech, 1:2000) CloneNo.2B3C4; anti-GFP tag antibody (50430-2-AP, Proteintech, 1:2000); β-actin antibody (sc-47778, Santa Cruz, 1:2000); anti-ADAM9 antibody (sc-377233, Santa Cruz, 1:1000); anti-ADAM17 antibody (sc-390859, Santa Cruz, 1:1000); mouse anti-rabbit IgG-HRP antibody (sc-2357, Santa Cruz, 1:5000); mouse anti-goat IgG-HRP antibody (sc-2354, Santa Cruz, 1:5000); m-IgGκ BP-HRP antibody (sc-516102, Santa Cruz, 1:5000); Alexa Fluor 568-conjugated rabbit anti-goat (A11079, Invitrogen, 1:500); Alexa Fluor 647-conjugated donkey anti-rabbit (A31573, Invitrogen, 1:500); Anti-NP antibody was kindly provided by the laboratory of Dr. Shuofeng Yuan (1:5000, Riva, L. et al. Discovery of SARS-CoV-2 antiviral drugs through large-scale compound repurposing. Nature 586, 113–119 (2020).; 3A2 antibody was kindly provided by the laboratory of Dr. Xin Ge

### Primary cell culture
**Primary culture of renal tubular epithelial cells.** Primary renal tubular cells were isolated from wildtype and *MMP14*[-/-] mice. Kidneys were manually minced and incubated with 1 mg/ml collagenase I in PBS with 0.5% BSA for 3 min at 37 °C. DMEM medium containing 10% FBS was

added to inactivate collagenase, and the mixture was passed first through a 70 μm filter and then through a 40 μm filter (BD Falcon, USA). Renal tubular epithelial cells were collected at the bottom of the 40 μm filter, seeded directly on DMEM culture plates containing 10% FBS medium, and cultured to confluence[67].

**Primary culture of lung epithelial cells.** Epithelial cells were isolated from mouse lung tissue by mechanical separation using a sterile sieve followed by discontinuous density gradient centrifugation using a Percoll (GE Healthcare Life Sciences, Chicago, IL, USA)[68]. Three Percoll solutions (25%, 40%, and 75%) were prepared, and cells from lung tissue were resuspended in the 40% solution, and the three different gradients were carefully placed in a 15 ml centrifuge tube (Corning, USA) in the order 25% – 40% – 75% from top to bottom. Cells were spun at $780 \times g$ for 20 min, with minimal acceleration and deceleration set. Epithelial cells were placed on the 25% and 40% Percoll solutions interface and then aspirated and washed several times with DMEM containing 10% FBS. Approximately $2 \times 10^6$ cells were inoculated in a 6-well plate coated with collagen.

**Airway epithelial culture.** Primary human bronchial epithelial cells (HBECs) were purchased from Sciencell (California, USA) and expanded using PneumaCult™-Ex Plus Medium (STEMCELL) in accordance to the manufacturer's protocols. The cell culture medium was changed every two days until confluence.

**Lentiviral infection and differentiation of HBECs.** Primary human bronchial epithelial cells were infected with shMMP14 lentivirus for 48 h and cultured with PneumaCult™-Ex Plus Medium supplemented with Puromycin (1 ug/ml) for 1 week. $4 \times 10^4$ cells were seeded at the apical chambers of 6.5 mm Transwell plates and maintained with PneumaCult™-Ex Plus medium. Two days after the incubation, the basolateral and apical chambers were removed and PneumaCult™-ALI maintenance medium (STEMCELL) was added to the basolateral chambers. The culture medium was replaced with new medium every two days and the ALI was differentiated in vitro for 28 days and subsequently subjected to SARS-CoV-2 infection.

**Primary culture of liver organoids.** Human liver organoids were kindly provided by Prof. Natalie Wong from The Chinese University of Hong Kong. Briefly, the non-tumoral liver tissue was obtained at the distant margin of resected liver cancer from patients who underwent curative surgery at Prince of Wales Hospital, Hong Kong. Informed consent was obtained from recruited patients with the study procedure approved by The Joint Chinese University of Hong Kong-New Territories East Cluster Clinical (CUHK-NTEC) Clinical Research Ethics Committee. Liver organoids were established from this tissue based on previously reported methods[69]. In brief, the tissue was mechanically dissociated by finely mincing with a scalpel. This was followed by enzymatic digestion with Collagenase (Worthington, USA) at 37 °C until cells were visibly released from the minced tissue. The reaction was then neutralized with FBS containing media and cells were passed through a 70-micron strainer (Corning, USA). The cell pellet was then resuspended in reduced growth factor matrigel (Corning, USA) and seeded in a prewarmed 12-well plate (Thermo Fisher, USA). Once the gel suspension has solidified, it is then topped up with expansion media containing defined growth factors as previously reported. Media was changed 1-2 times a week for maintenance. Upon reaching confluence, liver organoids were then either passaged with TrypLE express (Thermo Fisher, USA) for expansion or cryopreserved in freezing media (Thermo Fisher, USA) until future use. In some cases, TrypLE expression was used to isolate liver organs that were infected using shMMP14 lentivirus. Upon reaching confluence, liver organoids were isolated and incubated with the virus at 37° for 1 h. After washing, the inoculated organoids were re-embedded in Matrigel and cultured

in the medium. At the indicated times, liver organoids were harvested for the quantification of intracellular viral load.

**Human cardiac organoid culture.** Human cardiac organoids were purchased from Novoheart (Hong Kong) and cultured with Stem-Pro34 SFM (Invitrogen) medium supplemented with ascorbic acid (AA, 50 mg/mL; Sigma), 2 mM GlutaMAX-1 (Invitrogen), BMP4 (10 ng/mL), IWR-1 (5 mM; Enzo Life Sciences) and human recombinant activin-A (10 ng/mL; Invitrogen). The culture medium was replaced with the StemPro34 SFM medium with AA every two days[70]. Cardiac organoids were disassembled using TrypLE express upon reaching confluence, and infected with shMMP14 lentivirus for 24 h. Cardiac organoids were subsequently selected with Puromycin (0.5 ug/ml) for one week.

### Transfection and Protein Sample Collection
Cells grown on six-well plates were transfected with 1μg of the expression vector using Lipofectamine 3000 (Thermo Fisher, USA). 48 h after transfection, cells were washed twice with PBS (Gibco, USA) and incubated with serum-free DMEM at 37 °C for 12 h. The medium was collected and centrifuged at $16,000\,g$ for 3 min to remove detached cells. The conditioned medium was then concentrated to 40 μL using Amicon Ultra 0.5 mL Centrifugal Filters (Millipore, USA), followed by adding 10 μL of $5 \times$ loading buffer and boiling for 10 min. For cell lysis, the cells on the plates were washed twice with ice-cold PBS and lysed with RIPA buffer [150 mM NaCl, 1% Triton X-100, 0.5% deoxycholate, 0.1% SDS, 50 mM Tris (pH 7.5)] and 1 mM PMSF. Cell lysates were centrifuged at $16,000\,g$ for 15 min, and supernatants were collected for SDS/PAGE.

### Human plasma collection
Blood from young and elderly individuals was collected for EDTA-plasma isolation. 21 young (mean = 29.2 years; 13 males & 8 females) and 23 healthy elderly (mean = 80.1 years; 15 males & 8 females), mainly Asian Chinese, were recruited from Guanfu Cancer Hospital in Jinhua, Zhejiang Province, CHINA. They were all healthy, non-obese, with a BMI < 24. Informed consent was obtained from each individual. No participant compensation was provided. Blood was collected intravenously using a blood collection tube, left at room temperature for 30 min, and then centrifuged at 1000 x $g$ for 10 min at 4°. EDTA plasma was isolated, divided, and stored at −80° prior to analysis. The study protocol was conducted following the ethical guidelines of the hospital ethics committee, and the Zhejiang Provincial Health Committee approved the procedure. The protocol is in compliance with the Helsinki declaration.

### Adeno-associated virus (AAV) processing
pAAV-CMV-ACE2(1-17aa)-3xFLAG-ACE2(18-805aa)-WPRE, pAAV-CMV-ACE2(1-17aa)-3xFLAG-ACE2(18-740aa)-WPRE, and pAAV-CMV-ACE2(1-17aa)-3xFLAG-ACE2(18-706aa)-WPRE viruses were purchased from Obio Technologies Ltd. The vector plasmids are AAV vectors of serotype 2/9 under the control of the Cytomegalovirus (CMV) promoter, a well-documented AAV vector for broad-spectrum transgene expression in somatic cells. They were injected via the tail vein ($2.0 \times 10^{11}$ copies per mouse). At week 4, after AAV virus transduction, mice were infected intranasally with $1 \times 10^6$ PFUs of SARS-CoV-2 (WT). Two days post-infection, lungs were harvested for measurement of the virus titer. Mice were sacrificed at 2 dpi for virological and histopathological examination, organs, tissues and blood were collected for analysis, and harvested tissues were used for viral titration by RT-qPCR, and for histopathological and immunofluorescence analysis after fixation in 10% PBS-buffered formalin. For histopathological assessment, a semi-quantitative system was used to assess the relative degree of inflammation and tissue damage. Inflammation was scored as follows. 0, no inflammation; 1, perivascular cuff of inflammatory cells;

2, mild inflammation (extending to 25% of the lung); 3, moderate inflammation (25-50% of the lung); and 4, severe inflammation involving more than half of the lung. The examination of tissue sections was performed in a single-blind manner.

## Fab 3A2 treatment

C57BL/6J aged mice (14–16 months) were intraperitoneally administrated with Fab 3A2 antibody (10 mg/kg in 150 μl HBSS) the day before viral infection, while control mice received the vehicle alone. In the next day, mice were challenged by intranasal infection with SARS-CoV-2-N501Y, in a volume of 20 μL, $5 \times 10^5$ PFU/mice. The same dose of Fab 3A2 was given the day after infection. Mouse body weights were measured daily and status was observed. Two days after infection, lungs and trachea were harvested to measure viral titers by RT-qPCR and fixed with 10% PBS-buffered formaldehyde for histological and immunofluorescence analyses, and mouse plasma was collected for cytokine assay using the Bio-Plex Pro Mouse Cytokine Grp I Panel 23-plex (Bio-red) according to the manufacturer's instructions. The data were collected and analyzed by Bio-Plex 200 System (Luminex Corporation).

## Enzyme-linked immunosorbent assay

Mouse plasma was isolated from whole blood using EDTA anticoagulation tubes, and the blood was left at room temperature for 30 mins. Whole blood was centrifuged at $4,000 \, g$ for 20 min at 4 °C, and clarified plasma was collected. The concentrations of ACE2 and MT1-MMP in plasma samples were determined using ELISA kits and mouse ACE2 and MT1-MMP specific antibodies (SEB886Mu, SEC056Mu, Cloud-Clone, CHINA). The concentrations of ACE2 and MT1-MMP in human plasma samples were determined using human ACE2 (SEB886Hu, Cloud-Clone, CHINA) and human MT1-MMP (SEC056Hu, Cloud-Clone, CHINA) specific antibody-coated ELISA kits, respectively. The ELISA was performed according to the manufacturer's protocol.

## Western blotting analysis

Protein concentrations were measured using the BCA Protein Assay Kit. For SDS/PAGE, cell lysates containing the same amount of total proteins were loaded onto 4–20% precast Tris-HCl gels (Bio-Rad). Proteins were transferred to PVDF membranes (Millipore). All antibodies were diluted in TBST [50 mM Tris (pH 8.0), 150 mM NaCl, and 0.1% (vol/vol) Tween 20] containing 5% (wt/vol) BSA. secondary antibodies were horseradish peroxidase-conjugated mouse anti-goat, mouse anti-rabbit, or mouse IgG Fc binding protein (1:5,000, Santa Cruz, USA). Enhanced chemiluminescence (ECL) was detected using SuperSignal West Pico chemiluminescent substrate (Pierce, USA). Uncropped blots for the main and supplementary figures are provided at the end of supplementary information file.

## Immunoprecipitation

For immunoprecipitation with ectopically expressed proteins: 2 μg of anti-GFP polyclonal antibody or FLAG monoclonal antibody was incubated with 50 μl of Protein A/G magnetic beads for 30 min at room temperature by incubation with cell lysates from 100 mm plates transfected with ACE2-GFP and MT1-MMP-FLAG overnight at 4 °C. The protein-bound antibody-magnetic beads were washed twice with ice-cold RIPA buffer and ice-cold TSA [10 mM Tris (pH 8), 140 mM NaCl]. Purified proteins were eluted from the antibody-magnetic beads by boiling in a 2× Laemmli sample buffer for 5 min. The eluted protein samples were loaded onto 4–20% Tris-HCl gels and subjected to Western Blotting.

For immunoprecipitation of endogenous proteins: cells from two 150 mm culture dishes were lysed in pre-cooled RIPA buffer. A protein mixture at a concentration of 2 μg/μl was used for immunoprecipitation experiments. 50 μl of Protein A/G magnetic beads were mixed with 5 μg of MT1-MMP monoclonal antibody or ACE2 polyclonal antibody for 30 min at room temperature, followed by overnight mixing of all magnetic beads with the protein mixture at 4 °C. The protein-conjugated antibody-agarose pellets were washed twice with ice-cold RIPA buffer and ice-cold TSA [10 mM Tris (pH 8), 140 mM NaCl]. Purified proteins were eluted from the antibody-agarose beads by boiling in 2× Laemmli sample buffer for 5 min. The eluted protein samples were loaded onto 4–20% Tris-HCl gels and subjected to Western Blotting.

## Confocal and immunofluorescence microscopy analysis

HEK-293T cells were fixed in 4% paraformaldehyde in PBS containing 0.1% Triton X-100. After blocking with 3% bovine serum albumin (BSA) for 1 h at room temperature, the cells were incubated with anti-SARS-CoV-2 antibody (Sino Biological) and anti-ACE2 antibody (R&D System) at 4° overnight. Unbound antibodies were washed away five times with PBS. Positively stained cells were detected with Alexa Fluor 568 or Alexa Fluor 647 (Life Technology) conjugated secondary IgG (H+L) antibodies for 60 min at room temperature. After three washes with PBS, stained cells were mounted onto glass slides with SlowFade™ Gold Antifade Mountant with DAPI mounting medium (Thermo Fisher, USA) and examined by confocal imaging with Leica TCS SP8 (Leica).

## Real-time PCR analyses

Total RNA was isolated using the RNeasy kit (Beyotime, China) according to the manufacturer's method. The yield of RNA was determined by spectrophotometrically. RNA from each sample was reverse-transcribed into cDNA using a PrimeScript RT Master Mix Kit (Takara, Japan). PCR was performed using UltraSYBR Mixture (CWBIO, CHINA) for 1/10 of each RT sample under the following conditions: 95 °C for 10 min, followed by 35 replicate cycles: 95 °C (15 s) and 60 °C (60 s)). Gene expression was normalized with Gapdh mRNA levels. The results were analyzed by ViiA 7 Real-time PCR system with QuantStudio Software v1.6.1.

The primers used for PCR were: memACE2 (5'- TCCATTGGTCTTCTGCCATCC-3', 5'- AACGATCTCCCGCTTCATCTC-3'), GAPDH (5'- CATCACTGCCACCCAGAAGACTG-3', 5'- ATGCCAGTGAGCTTCCCGTTCAG-3'). The primer sequences for generation ACE2 mutants by site-directed mutagenesis were listed in Supplementary Table 1.

## In vitro MT1-MMP cleavage assay

The catalytic domain of recombinant MT1-MMP was kindly provided by Dr. Xin Ge from University of Texas Health Science Center at Houston. The recombinant Human ACE-2 (933-ZN) were purchased from R&D systems. SARS-CoV-2 WT S1-His Recombinant Protein (40591-V08H), SARS-CoV-2 (Delta) S1-His Recombinant Protein (40591-V08H23) and SARS-CoV-2 B.1.1.529 (Omicron) S1-His Recombinant Protein (40591-V08H41) were purchased from Sino Biological (CHINA). Human ACE-2 protein (Gln18-Ser740) with a C-terminal 10-His tag was purified from NS0 cells. They were incubated in assay buffer (50 mM Tris-HCl pH 7.5, 150 mM NaCl, 5 mM CaCl2 and 0.025% Brij35) at 37 °C for 6 h. Protein mixtures were subjected to Western blotting analyses.

## C-terminal sequencing of ACE2 fragment

10 ug of ACE2 and MT1-MMP recombinant proteins were incubated overnight at 37 °C in the assay buffer (50 mM Tris-HCl pH 7.5, 150 mM NaCl, and 5 mM CaCl2), followed by separation of the recombinant proteins using SDS-PAGE. C-terminal sequencing analysis was performed by Bio-Tech Pack Technology Company Ltd. (China) using the Q Exactive™ Plus Hybrid Quadrupole-Orbitrap™ Mass Spectrometer (Thermo Scientific, USA).

## Generation and infection of sh*MT1-MMP* lentivirus and SARS-CoV-2 virus pseudotypes

Briefly, $3 \times 10^6$ HEK293T cells were cultured on 10 cm plates with DMEM supplemented with 10% FBS, penicillin/streptomycin, and L-glutamine. In the next day, cells were transfected according to the manufacturer's instructions using Lipofectamine 3000 transfection reagent (Invitrogen, USA) with sh*MMP14*, psPAX2 and pMD2.G or psPAX2, pLenti-Luc and SARS-CoV-2 S plasmids for cotransfection. Supernatants were collected at 48 and 72 h post-transfection, passed through a 0.45 μm filter and subsequently centrifuged for 2 h at 70,000 x *g* using a Type 70 Ti Fixed-Angle Titanium Rotor (BECKMAN, USA) at 4 °C. Virus particles were resuspended and then aliquoted and stored at −80 °C. To infect target cells with SARS-CoV-2 virus pseudotypes, cells were seeded into 96-well plates, transfected with the indicated plasmids or siRNAs overnight, and then infected with SARS-CoV-2 virus pseudotypes for 72 h. Cells were washed three times, followed by lysis and detection of luciferase intensity using the Bright-Lumi™ Firefly Fluorosceinase Reporter Gene Assay Kit (Beyotime, CHINA).

## Authentic SARS-CoV-2 infection of human cells

SARS-CoV-2 HKU-001a strain (GenBank accession number: MT230904), SARS-CoV-2 Delta variant (GISAID: EPI_ISL_9681416) and SARS-CoV-2 Omicron variant BA.1 (GISAID: EPI_ISL_7357684) were isolated from nasopharyngeal aspirate specimens from laboratory-confirmed patients with COVID-19 in Hong Kong. The initial generation of virus was plaque purified and massively amplified using VeroE6 cells overexpressing TMPRSS2, and the virus was stored at −80 °C. All authentic SARS-CoV-2 infection experiments were performed using these early passages of SARS-CoV-2 to ensure the consistency of our experiments. The siRNA was transfected into Caco2 cells using Lipofectamine RNAiMAX, and ells were infected using 0.1 MOI at 24 h post-transfection. On the third day post-infection, total RNA was collected from the siRNA-transfected cells for detection of SARS-CoV-2 viral copy by real time-quantitative RT-PCR detecting the 5-ACAGGTACGTTAATAGCTAGCGT -3 locus of SARS-CoV-2. All experiments involving live SARS-CoV-2 followed the approved standard operating procedures for biosafety level 3 facilities at the University of Hong Kong, as previously described[71].

## Analyses of single cell transcriptome in human tissues

*ACE2* and *MMP14* expression were analyzed using a recently published single-cell mRNA sequencing dataset obtained from human lung tissues. The data processed in the form of Seurat objects was downloaded via FigShare: (https://doi.org/10.6084/m9.figshare.12436517, https://doi.org/10.6084/m9.figshare.13200278, and https://doi.org/10.6084/m9.figshare.14938755).

Analysis was performed using R 3.6.0 and RStudio 1.1.463. dplyr[1] 0.8.4, plyr[2] 1.8.4, and stringr[3] 1.4.0 packages were used for data manipulation. Raw count data were normalized using DESeq2[4] 1.26 with default settings and filtered to keep only genes with more than ten counts across all samples. Violin plots, boxplots, and scatter plots were generated using ggplot2[5] 3.3.2, ggbeeswarm[6] 0.6.0, and ggpubr[7] 0.2.5. Data were subset to include only cells from lung tissue. Uniform flow approximation projection (UMAP) visualizations were annotated with metadata for a wide range of cell types using the "interval" data field and interrogated for expression data for individual genes (ACE2 and MMP14).

## Statistics and reproducibility

Each experiment was independently performed for at least three times. Animal experiments involved at least three independent and randomly chosen mice at comparable developmental stages and none of the samples were excluded from analyses. Sample size was determined from the power of the statistical test performed and was increased in accordance to the statistical variation. The statistical differences were determined using one-way analysis of variance (ANOVA) followed by Tukey's post hoc test, Mann–Whitney U test or student's *t* test on GraphPad Prizm 8.0 software. All values are expressed as means ± s.e.m. along with the number of individual mice/samples analyzed (n). All data meet the normal distribution. *P*-value of < 0.05 is accepted as statistically significant.

## Reporting summary

Further information on research design is available in the Nature Portfolio Reporting Summary linked to this article.

## Data availability

All relevant data supporting the key findings of this study are available within the article and its supplementary information files. The datasets analyzed during the current study are available in FigShare (https://doi.org/10.6084/m9.figshare.12436517, https://doi.org/10.6084/m9.figshare.13200278, and https://doi.org/10.6084/m9.figshare.14938755) and Human Protein Atlas. Source data are provided with this paper.

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

## Acknowledgements

The presented work was kindly supported by General Research Fund (12101019 and 12102020; W.H.L.X.), Research Impact Fund (R4017-18; W.N.), Health and Medical Research Fund (06170056 and 08793626; W.H.L.X.), National Natural Science Fund (81802838; W.H.L.X.) and Guangdong Natural Science Foundation (2021A1515011128; W.H.L.X.).

## Author contributions

G.X., Y.S. and W.H.L.X. conceived the project. G.X. with the help from W.J., G.S., and Z.Y. performed most of the experiments. Y.S., C.J and C.J.P. performed the virus-related experiments. H.J., W.S.K.K., H.J., Y.Z.W., W.S., W.Z., G.X., W.N. and Z.Z. provided the research materials and collected the samples. K.H.Y., L.A., C.K.M. and B.Z.X. provided suggestions on the research study. G.X., C.J. and Y.S. analyzed the data with bioinformatic analyses. G.X., A.P. and W.H.L.X. wrote the paper.

## Competing interests

The authors declare no competing interests.

## Ethics approval

The criteria of authorship was set in accordance to the Global Code of Conduct for Research.
