## [Peer Review File · Nature Communications]

Control of SARS-CoV-2 infection by MT1-MMP-mediated shedding of ACE2Editorial Note: This manuscript has been previously reviewed at another journal that is not operating a transparent peer review scheme. This document only contains reviewer comments and rebuttal letters for versions considered at Nature Communications.

Reviewers' Comments:

Reviewer #1:

Remarks to the Author:

The authors have already addressed my previous concerns.

Reviewer #2:

Remarks to the Author:

The revised manuscript have improved significantly. The author have addressed most of the concerns raised. This reviewer still have several questions need to be answered before publication.

1, supplementary Figure 3a, the cleavage experiment of S707A shows it could still be degraded, which is weird. Therefore, the cleavage site still needs to be further clarified. The authors could use the purified rACE2 (1-706) to check whether it could promote the infection to eliminate the concern.

2. In figure 3c and the last paragraph of page 6, Caco2 as a SARS-CoV-2 permissive cell line were used. In supporting the conclusion that rACE2 itself facilitated the virus entry, a cell line that don't have endogenous ACE2 expression should be used.

3. The last paragraph of page 7: How to explain that "this stimulating effect was further exacerbated with rACE2 digested by catalytic domain of MT1-MMP"? Furthermore, the detailed information of the rACE2 used here is lacking. A rACE2 at the same size with solACE2 digested by MT1-MMP, and also a cell line that doesn't have endogenous ACE2 expression should be used here to support the conclusion

4. In previous work, Yamamoto et, al presented that, Metalloproteinase facilitated cell entry by S protein digestion requires the furin-cleavage site. However, in Figure S10b, seems the recombinant S protein have the furin cleavage site mutated.

(Yamamoto M, Gohda J, Kobayashi A, Tomita K, Hirayama Y, Koshikawa N, et al.

Metalloproteinase-dependent and TMPRSS2-independent cell surface entry pathway of SARS-CoV-2 requires the furin-cleavage site and the S2 domain of spike protein. bioRxiv.2021:2021.12.14.472513. doi: 10.1101/2021.12.14.472513.)

Reviewer #1

The authors have already addressed my previous concerns.

Answer: Thank you very much for your kind efforts on improving our manuscript

Reviewer #2

The revised manuscript have improved significantly. The author have addressed most of the concerns raised. This reviewer still have several questions need to be answered before publication.

Answer: Thank you very much for your encouraging comments on our manuscript.

1. supplementary Figure 3a, the cleavage experiment of S707A shows it could still be degraded, which is weird. Therefore, the cleavage site still needs to be further clarified. The authors could use the purified rACE2 (1-706) to check whether it could promote the infection to eliminate the concern.

Answer: Thank you very much for your excellent suggestions. To further confirm the identified cleavage site, the truncated fragment of ACE2 resulted from MT1-MMP cleavage in the *in vitro* cleavage assay was sequenced by mass spectrometry (MS) coupled with tandem MS/MS, revealing that MT1-MMP indeed cleaved C-terminus of ACE2 at M706-S (**Supplementary Fig. 4**).

To investigate whether solACE2-mediated entry of SARS-CoV-2 depends on memACE2, we synthesized recombinant ACE2 comprising M1-M706 (rACE2₁₋₇₀₆) that corresponded to solACE2 released from MT1-MMP cleavage site. A549 and HEK293T cells, two well-known SARS-CoV-2 nonpermissive cell lines due to lack of ACE2 expression, were inoculated with rACE2₁₋₇₀₆ and then subjected to SARS-CoV-2 infection. Without the presence of rACE2₁₋₇₀₆, neither viral antigens nor gene copies were detected in both cell lines (**Fig. 4a-c**). In contrast, the addition of rACE2₁₋₇₀₆ in a dose-dependent manner promoted the infection of SARS-CoV-2 in both cells (**Fig. 4a-c**). These results suggest that solACE2 released by MT1-MMP facilitates SARS-CoV-2 infection in a memACE2-independent manner.

To investigate whether solACE2 released by MT1-MMP facilitates SARS-CoV-2 entry *in vivo*, we designed an adeno-associated viral vector under the control of CMV promoter, a potent promoter commonly used for mediating the expression of transgenes, to ectopically and systemically express either soluble form of human ACE2 comprising M1-M706 (solACE2₁₋₇₀₆) that corresponded to solACE2 released from MT1-MMP cleavage site or human ACE2 with the deletion of transmembrane and cytosolic domains (solACE2₁₋₇₄₀) or full length membrane form of ACE2 (ACE2_{FL}) as a positive control or blank control in wild-type C57BL/6 mice that were naturally resistant to infection with wild-type SARS-CoV-2. Mice were firstly transduced with adenoviral particles for 14 days for facilitating stable expression of different forms of solACE2, followed by challenge with wild-type SARS-CoV-2 for another 2 days (**Fig. 4d**). To investigate whether solACE2 facilitates SARS-CoV-2 entry, we detected the expression of human ACE2 and SARS-CoV-2 Nucleocapsid Protein (NP) by immunofluorescent staining (**Fig. 4e**). Unlike control mice that were insusceptible to SARS-CoV-2 infection even with high infectious doses, viral antigens could be detected in lung sections of mice expressing solACE2₁₋

solACE2₁₋₇₄₀ / ACE2_{FL} upon intranasal infection of SARS-CoV-2 (**Fig. 4e**). SARS-CoV-2 NP was highly colocalized with human ACE2 in alveolar linings of mice expressing solACE2₁₋₇₀₆ / solACE2₁₋₇₄₀ (**Fig. 4e**), indicative of the internalization of human solACE2/SARS-CoV-2 viral complex in alveolar cells. Importantly, the level of this viral complex in the lungs of solACE2₁₋₇₀₆-expressing mice, which was comparable to that observed in mice expressing ACE2_{FL}, was considerably higher than that in solACE2₁₋₇₄₀-expressing mice (**Fig. 4e; supplementary Fig. 8a**). Concordant with the antigen detection, viral gene copies were significantly elevated in lungs of solACE2₁₋₇₀₆-expressing mice when compared to solACE2₁₋₇₄₀-expressing mice (**Fig. 4f**). In addition, solACE2₁₋₇₀₆-expressing mice exhibited the similar degree of tissue damages (e.g. perivascular to interstitial inflammatory cell infiltrates, necrotic cell debris, and alveolar edema) and lung inflammatory responses as those in mice expressing ACE2_{FL} at 2 dpi (**Fig. 4g-h; supplementary Fig. 8b**). These data suggest that solACE2 released by MT1-MMP is as important as full-length memACE2 in mediating SARS-CoV-2 infection *in vivo*. The tissue damage resulted from viral infection in our experimental model suggests that solACE-2-directed infections are physiologically relevant.

2. In figure 3c and the last paragraph of page 6, Caco2 as a SARS-CoV-2 permissive cell line were used. In supporting the conclusion that rACE2 itself facilitated the virus entry, a cell line that don't have endogenous ACE2 expression should be used.

Answer: As mentioned above, we demonstrated that rACE2₁₋₇₀₆ facilitated the virus entry in A549 and HEK293T cells, two well-known SARS-CoV-2 nonpermissive cell lines (**Fig. 4a-c**). In addition, the expression of human solACE2₁₋₇₀₆ also enabled SARS-CoV-2 infection in naturally insusceptible mice (**Fig. 4d-h**).

3. The last paragraph of page 7: How to explain that "this stimulating effect was further exacerbated with rACE2 digested by catalytic domain of MT1-MMP"? Furthermore, the detailed information of the rACE2 used here is lacking. A rACE2 at the same size with solACE2 digested by MT1-MMP, and also a cell line that doesn't have endogenous ACE2 expression should be used here to support the conclusion

Answer: Commercially available rACE2 corresponds to human ACE2 with the deletion of transmembrane and cytosolic domains (solACE2₁₋₇₄₀). Our *in vitro* assay using recombinant proteins showed that solACE2₁₋₇₀₆ was generated upon the cleavage of rACE2 by MT1-MMP (**Fig. 1e, Supplementary Fig. 4**). This physiologically relevant fragment of ACE2 (solACE2₁₋₇₀₆) was more efficient than rACE2 (solACE2₁₋₇₄₀) in promoting SARS-CoV-2 infection in cell lines (**Fig. 3h**) and naturally resistant mice (**Fig. 4d-h**).

4. In previous work, Yamamoto et, al presented that, Metalloproteinase facilitated cell entry by S protein digestion requires the furin-cleavage site. However, in Figure S10b, seems the

recombinant S protein have the furin cleavage site mutated.

Answer: Thank you for your suggestions. In the work by Yamamoto et al, they investigated the effect of metalloproteinases on virus entry using broad-spectrum MMP inhibitors that can target both MMPs and ADAMs families. Therefore, further investigations would be required to identify the key MMP involved in SARS-CoV-2 entry and investigate its underlying mechanism. In our current study, we have delineated the detailed mechanism regulating the release of solACE2 by identifying the major host protease responsible for ACE2 shedding. By comparing with other well-known proteases (e.g. MMP2, MMP9, MMP13, ADAM9 and ADAM17) involved in ectodomain-shedding of cell surface proteins, only MT1-MMP was found to efficiently cleave memACE2 and release functionally active solACE2₁₈₋₇₀₆. This ACE2 fragment is indeed the dominant solACE detected in both human and mouse tissues, and can be largely depleted by inhibition of MT1-MMP activities. Given the facts that adenoviral transduction of human solACE2₁₈₋₇₀₆ enables SARS-CoV-2 infection in non-permissive cell lines (**Fig. 4a-c**) and naturally resistant C57BL/6 mice (**Fig. 4d-h**) as well as blockage of ACE2 shedding by MT1-MMP effectively attenuates SARS-CoV-2 entry in *ex vivo* and *in vivo* (**Fig. 6-7**), we are the first to provide the *in vivo* evidence showing the contribution of ACE2 shedding to the etiology of COVID-19.

In original Figure S10b, we used recombinant S2 proteins for the *in vitro* assay. To further investigate whether MT1-MMP cleaves S protein of SARS-CoV-2, we incubated the recombinant full-length S proteins (S1+S2) with the catalytic domain of MT1-MMP *in vitro* (**figure below**). Unlike furin, which could digest S proteins, the addition of MT1-MMP did not result in significant changes in the levels of S protein and its degraded fragments. This data suggests that MT1-MMP unlikely cleaves S proteins. Even though MMP cleavage-fusion activation of spike proteins has been demonstrated to promote SARS-CoV-2 entry, our study suggests that other MMPs, but not MT1-MMP, may be involved in this cleavage process. The differential roles of different MMPs in SARS-CoV-2 infection certainly warrants further investigation in the future.

Reviewers' Comments:

Reviewer #2:

Remarks to the Author:

The revised manuscript have improved significantly. The author have addressed the concerns raised. I recommend the publicaiton of the manuscript.